# Instance Based Approximations to Profile Maximum Likelihood

**Nima Anari**
Stanford University
anari@stanford.edu

**Moses Charikar**
Stanford University
moses@cs.stanford.edu

**Kirankumar Shiragur**
Stanford University
shiragur@stanford.edu

**Aaron Sidford**
Stanford University
sidford@stanford.edu

## Abstract

In this paper we provide a new efficient algorithm for approximately computing the profile maximum likelihood (PML) distribution, a prominent quantity in symmetric property estimation. We provide an algorithm which matches the previous best known efficient algorithms for computing approximate PML distributions and improves when the number of distinct observed frequencies in the given instance is small. We achieve this result by exploiting new sparsity structure in approximate PML distributions and providing a new matrix rounding algorithm, of independent interest. Leveraging this result, we obtain the first provable computationally efficient implementation of PseudoPML, a general framework for estimating a broad class of symmetric properties. Additionally, we obtain efficient PML-based estimators for distributions with small profile entropy, a natural instance-based complexity measure. Further, we provide a simpler and more practical PseudoPML implementation that matches the best-known theoretical guarantees of such an estimator and evaluate this method empirically.

## 1 Introduction

In this paper we consider the fundamental problem of *symmetric property estimation*: given access to $n$ i.i.d. samples from an unknown distribution, estimate the value of a given symmetric property (i.e. one invariant to label permutation). This is an incredibly well-studied problem with numerous applications [Cha84, BF93, CCG+12, TE87, Für05, KLR99, PBG+01, DS13, RCS+09, GTPB07, HHRB01] and proposed property-specific estimators, e.g. for support [VV11b, WY15], support coverage [ZVV+16, OSW16], entropy [VV11b, WY16a, JVHW15], and distance to uniformity [VV11a, JHW16].

However, in a striking recent line of work it was shown that there is *a universal approach* to achieving sample optimal[1] estimators for a broad class of symmetric properties, including those above. [ADOS16] showed that the value of the property on a distribution that (approximately) maximizes the likelihood of the observed profile (i.e. multiset of observed frequencies) is an optimal estimator up to accuracy[2] $\epsilon \gg n^{-1/4}$. Further, [ACSS20] which in turn built on [ADOS16, CSS19a], provided a polynomial time algorithm to compute an $\exp(-O(\sqrt{n} \log n))$-approximate profile maximum likelihood distribution (PML). Together, these results yield efficient sample optimal estimators for various symmetric properties up to accuracy $\epsilon \gg n^{-1/4}$.

Despite this seemingly complete picture of the complexity of PML, recent work has shown that there is value in obtaining improved approximate PML distributions. In [CSS19b, HO19] it was shown that variants of PML called *PseudoPML* and *truncated PML* respectively, which compute an approximate PML distribution on a subset of the coordinates, yield sample optimal estimators in broader error regime for a wide range of symmetric properties. Further, in [HO20] an instance dependent quantity known as *profile entropy* was shown to govern the accuracy achievable by PML and their analysis holds for all symmetric properties with no additional assumption on the structure of the property. Additionally, in [HS20] it was shown that PML distributions yield a sample optimal universal estimator up to error $\epsilon \gg n^{-1/3}$ for a broad class of symmetric properties. However, the inability to obtain approximate PML distributions of approximation error better than $\exp(-O(\sqrt{n}\log n))$ has limited the provably efficient implementation of these methods.

In this paper we enable many of these applications by providing improved efficient approximations to PML distributions. Our main theoretical contribution is a polynomial time algorithm that computes an $\exp(-O(k\log n))$-approximate PML distribution where $k$ is the number of distinct observed frequencies. As $k$ is always upper bounded by $\sqrt{n}$, our work generalizes the previous best known result from [ACSS20] that computed an $\exp(-O(\sqrt{n}\log n))$-approximate PML. Leveraging this result, our work provides the first provably efficient implementation of PseudoPML. Further, our work also yields the first provably efficient estimator for profile entropy and efficient estimators with instance-based high-accuracy guarantees via profile entropy. We obtain our approximate PML result by leveraging interesting sparsity structure in convex relaxations of PML [ACSS20, CSS19a] and additionally provide a novel matrix rounding algorithm that we believe is of independent interest.

Finally, beyond the above theoretical results we provide a simplified instantiation of these results that is sufficient for implementing PseudoPML. We believe this result is a key step towards practical PseudoPML. We provide preliminary experiments in which we perform entropy estimation using the PseudoPML approach implemented using our simpler rounding algorithm. Our results match other state-of-the-art estimators for entropy, some of which are property specific.

**Notation and basic definitions:**  Throughout this paper we assume to receive a sequence of $n$ independent samples from an underlying distribution $\mathbf{p} \in \Delta^{\mathcal{D}}$, where $\mathcal{D}$ is a domain of elements and $\Delta^{\mathcal{D}}$ is the set of all discrete distributions supported on this domain. We let $[a, b]$ and $[a, b]_{\mathbb{R}}$ denote the interval of integers and reals $\geq a$ and $\leq b$ respectively, so $\Delta^{\mathcal{D}} \stackrel{\text{def}}{=} \{\mathbf{q} \in [0, 1]_{\mathbb{R}}^{\mathcal{D}} | \|q\|_1 = 1\}$. Let $\mathcal{D}^n$ be the set of all length $n$ sequences and $y^n \in \mathcal{D}^n$ be one such sequence with $y_i^n$ denoting its $i$th element. Let $\mathbf{f}(y^n, x) \stackrel{\text{def}}{=} |\{i \in [n] \mid y_i^n = x\}|$ and $\mathbf{p}_x$ be the frequency and probability of $x \in \mathcal{D}$ respectively. For a sequence $y^n \in \mathcal{D}^n$, let $\mathbf{M} = \{\mathbf{f}(y^n, x)\}_{x \in \mathcal{D}} \backslash \{0\}$ be the set of all its non-zero distinct frequencies and $\mathbf{m}_1, \mathbf{m}_2, \ldots, \mathbf{m}_{|\mathbf{M}|}$ be these distinct frequencies. The *profile* of a sequence $y^n$ denoted $\phi = \Phi(y^n)$ is a vector in $\mathbb{Z}_+^{|\mathbf{M}|}$, where $\phi_j \stackrel{\text{def}}{=} |\{x \in \mathcal{D} \mid \mathbf{f}(y^n, x) = \mathbf{m}_j\}|$ is the number of domain elements with frequency $\mathbf{m}_j$. We call $n$ the length of profile $\phi$ and let $\Phi^n$ denote the set of all profiles of length $n$. The probability of observing sequence $y^n$ and profile $\phi$ with respect to a distribution $\mathbf{p}$ are as follows,

$$\mathbb{P}(\mathbf{p}, y^n) = \prod_{x \in \mathcal{D}} \mathbf{p}_x^{\mathbf{f}(y^n, x)} \quad \text{and} \quad \mathbb{P}(\mathbf{p}, \phi) = \sum_{\{y^n \in \mathcal{D}^n \mid \Phi(y^n) = \phi\}} \mathbb{P}(\mathbf{p}, y^n) \,.$$

For a profile $\phi \in \Phi^n$, $\mathbf{p}_\phi$ is a *profile maximum likelihood* (PML) distribution if $\mathbf{p}_\phi \in \arg\max_{\mathbf{p} \in \Delta^{\mathcal{D}}} \mathbb{P}(\mathbf{p}, \phi)$. Further, a distribution $\mathbf{p}_\phi^\beta$ is a $\beta$-*approximate PML* distribution if $\mathbb{P}(\mathbf{p}_\phi^\beta, \phi) \geq \beta \cdot \mathbb{P}(\mathbf{p}_\phi, \phi)$.

For a distribution $\mathbf{p}$ and $n$, let $\mathbf{X}$ be a random variable that takes value $\phi \in \Phi^n$ with probability $\Pr(\mathbf{p}, \phi)$. The distribution of $\mathbf{X}$ depends only on $\mathbf{p}$ and $n$ and we call $H(\mathbf{X})$ (entropy of $\mathbf{X}$) the *profile entropy* with respect to $(\mathbf{p}, n)$ and denote it by $H(\Phi^n, \mathbf{p})$.

We use $\widetilde{O}(\cdot), \widetilde{\Omega}(\cdot)$ notation to hide all polylogarithmic factors in $n$ and $N$.

**Paper organization:**  In Section 2 we formally state our results. In Section 3, we provide the convex relaxation [CSS19a, ACSS20] for the PML objective. Using this convex relaxation, in Section 4 we state our algorithm that computes an $\exp(-O(k\log n))$-approximate PML and sketch its proof. Finally, in Section 5, we provide a simpler algorithm that provably implements the PseudoPML approach; we implement this algorithm and provide experiments in the same section. Due to space constraints, we defer most of the proofs to appendix.

## 2 Results

Here we provide the main results of our paper on computing approximations to PML where the approximation quality depends on the number of distinct frequencies, as well as efficiently implementing results on profile entropy and PseudoPML.

**Distinct frequencies:** Our main approximate PML result is the following.

**Theorem 2.1** (Approximate PML). *There is an algorithm that given a profile $\phi \in \Phi^n$ with $k$ distinct frequencies, computes an $\exp\left(-O(k \log n)\right)$-approximate PML distribution in time polynomial in $n$.*

Our result generalizes [ACSS20] which computes an $\exp(-O(\sqrt{n} \log n))$-approximate PML. Through [ADOS16] our result also provides efficient optimal estimators for class of symmetric properties when $\epsilon \gg n^{-1/4}$. Further, for distributions that with high probability output a profile with $O(n^{1/3})$ distinct frequencies, through [HS20] our algorithm enables efficient optimal estimators for the same class of properties when $\epsilon \gg n^{-1/3}$. In Section 4 we provide a proof sketch for the above theorem and defer all the proof details to Appendix A.

**Profile entropy:** One key application of our instance-based, i.e. distinct-frequency-based, approximation algorithm is the efficient implementation of the following approximate PML version of the profile entropy result from [HO20].[3] See Section 1 for the definition of profile entropy.

**Lemma 2.2** (Theorem 3 in [HO20]). *Let $f$ be a symmetric property. For any $\boldsymbol{p} \in \Delta^{\mathcal{D}}$ and a profile $\phi \sim \boldsymbol{p}$ of length $n$ with $k$ distinct frequencies, with probability at least $1 - O(1/\sqrt{n})$,*

$$|f(\boldsymbol{p}) - f(\boldsymbol{p}_\phi^\beta)| \leq 2\epsilon_f \left( \frac{\widetilde{\Omega}(n)}{\lceil H(\Phi^n, \boldsymbol{p}) \rceil} \right) ,$$

*where $\boldsymbol{p}_\phi^\beta$ is any $\beta$-approximate PML distribution for $\beta > \exp(-O(k \log n))$ and $\epsilon_f(n)$ is the smallest error that can be achieved by any estimator with sample size $n$ and success proability[4] $9/10$*

As the above result requires an $\exp(-O(k \log n))$-approximate PML, our Theorem 2.1 immediately provides an efficient implementation of it. Lemma 2.2 holds for any symmetric property with no additional assumptions on the structure. Further, it trivially implies a weaker result in [ADOS16] where $\lceil H(\Phi^n, \boldsymbol{p}) \rceil$ is replaced by $\sqrt{n}$. For further details and motivation, see [HO20].

**PseudoPML:** Our approximate PML algorithm also enables the efficient implementation of PseudoPML [CSS19b, HO19]. Using PseudoPML, the authors in [CSS19b, HO19] provide a general estimation framework that is sample optimal for many properties in wider parameter regimes than the previous universal approaches. At a high level, in this framework, the samples are split into two parts based on the element frequencies. The empirical estimate is used for the first part and for the second part, they compute the estimate corresponding to approximate PML. To efficiently implement the approach of PseudoPML required efficient algorithms with either strong or instance dependent approximation guarantees and our result (Theorem 2.1) achieves the later. We first state a lemma that relates the approximate PML computation to the PseudoPML.

**Lemma 2.3** (PseudoPML). *Let $\phi \in \Phi^n$ be a profile with $k$ distinct frequencies and $\ell, u \in [0, 1]$. If there exists an algorithm that runs in time $T(n, k, u, \ell)$ and returns a distribution $\boldsymbol{p}'$ such that*

$$\mathbb{P}(\boldsymbol{p}', \phi) \geq \exp\left(-O((u - \ell)n \log n + k \log n)\right) \max_{\boldsymbol{q} \in \Delta^{\mathcal{D}}_{[\ell, u]}} \mathbb{P}(\boldsymbol{q}, \phi) , \tag{1}$$

*where $\Delta^{\mathcal{D}}_{[\ell, u]} \stackrel{\text{def}}{=} \{\boldsymbol{p} \in \Delta^{\mathcal{D}} | \boldsymbol{p}_x \in [\ell, u] \ \forall x \in \mathcal{D}\}$. Then we can implement the PseudoPML approach with the following guarantees,*

- *For entropy, when error parameter $\epsilon > \Omega\left(\frac{\log N}{N^{1-\alpha}}\right)$ for any constant $\alpha > 0$, the estimator is sample complexity optimal and runs in $T(n, O(\log n), O(\log n/n), 1/\text{poly}(n))$ time.*

- *For distance to uniformity, when $\epsilon > \Omega\left(\frac{1}{N^{1-\alpha}}\right)$ for any constant $\alpha > 0$, the estimator is sample complexity optimal and runs in $T(n, \widetilde{O}(1/\epsilon), O(1/N), \Omega(1/N))$ time.*

The proof of the lemma is divided into two main steps. In the first step, we relate (1) to conditions considered in PseudoPML literature. In the second step, we leverage this relationship and the analysis in [CSS19b, HO19] to obtain the result. See Appendix B.3 for the proof of the lemma and other details. As discussed in [CSS19b, HO19], the above results are interesting because we have a general framework (PseudoPML approach) that is sample optimal in a broad range of non-trivial estimation settings; for instance when $\epsilon < \frac{\log N}{N}$ for entropy and $\epsilon < \frac{1}{N^C}$ for distance to uniformity where $C > 0$ is a constant, we know that the empirical estimate is optimal.

As our approximate PML algorithm (Theorem 2.1) runs in time polynomial in $n$ (for all values of $k$) and returns a distribution that satisfies the condition of the above lemma; we immediately obtain an efficient implementation of the results in Lemma 2.3. However for practical purposes, we present a simpler and faster algorithm that outputs a distribution which suffices for the application of PseudoPML. We summarize this result in the following theorem.

**Theorem 2.4** (Efficient PseudoPML). *There exists an algorithm that implements Lemma 2.3 in time $T(n, k, u, \ell) = \widetilde{O}(n\, k^{\omega - 1} \log \frac{u}{\ell})$, where $\omega$ is the matrix multiplication constant. Consequently, this provides estimators for entropy and distance to uniformity in time $\widetilde{O}(n)$ and $\widetilde{O}(n/\epsilon^{\omega-1})$ under their respective error parameter restrictions.*

See Section 5 for a description of the algorithm and proof sketch. The running time in the above result involves: solving a convex program, $n/k$ number of linear system solves of $k \times k$ matrices and other low order terms for the remaining steps. In our implementation we use CVX[GB14] with package CVXQUAD[FSP17] to solve the convex program. We use couple of heuristics to make our algorithm more practical and we discuss them in Appendix B.4.

## 2.1 Related work

PML was introduced by [OSS+04]. Since then, many heuristic approaches [OSS+04, ADM+10, PJW17, Von12, Von14] have been proposed to compute an approximate PML distribution. Recent work of [CSS19a] gave the first provably efficient algorithm to compute a non-trivial $\exp(-O(n^{2/3} \log n))$-approximate PML distribution. The proof of this result is broadly divided into three steps. In the first step, the authors in [CSS19a] provide a convex program that approximates the probability of a profile for a fixed distribution. In the second step, they perform minor modifications to this convex program to reformulate it as instead maximizing over all distributions while maintaining the convexity of the optimization problem. The feasible solutions to the modified convex program represent fractional distributions and in the third step, a rounding algorithm is applied to obtain a valid distribution. The approximation quality of this approach is governed by the first and last step and [CSS19a] showed a loss of $\exp(-O(n^{2/3} \log n))$ for each and thereby obtained $\exp(-O(n^{2/3} \log n))$-approximate PML distribution. In follow up work, [ACSS20] improved the analysis for the first step and then provided a better rounding algorithm in the third step to output an $\exp(-O(\sqrt{n} \log n))$-approximate PML distribution. The authors in [ACSS20] showed that the convex program considered in the first step by [CSS19a] approximates the probability of a profile for a fixed distribution up to accuracy $\exp(-O(k \log n))$, where $k$ is the number of distinct observed frequencies in the profile. However they incurred a loss of $\exp(-O(\sqrt{n} \log n))$ in the rounding step; thus returning an $\exp(-O(\sqrt{n} \log n))$ PML distribution. To prove these results, [CSS19a] used a combinatorial view of the PML problem while [ACSS20] analyzed the Bethe/Sinkhorn approximation to the permanent [Von12, Von14].

Leveraging the connection between PML and symmetric property estimation, [CSS19a] and [ACSS20] gave efficient optimal universal estimators for various symmetric properties when $\epsilon \gg n^{-1/6}$ and $\epsilon \gg n^{-1/4}$ respectively. The broad applicability of PML in property testing and to estimate other symmetric properties was later studied in [HO19]. [HS20] showed interesting continuity properties of PML distributions and proved their optimality for sorted $\ell_1$ distance and other symmetric properties when $\epsilon \gg n^{-1/3}$; no efficient version of this result is known yet.

There have been other approaches for designing universal estimators, e.g. [VV11b] based on [ET76], [HJW18] based on local moment matching, and variants of PML by [CSS19b, HO19] that weakly

depend on the property. Optimal sample complexities for estimating many symmetric properties were also obtained by constructing property specific estimators, e.g. sorted $\ell_1$ distance [VV11a, HJW18], Renyi entropy [AOST14, AOST17], KL divergence [BZLV16, HJW16] and others.

## 2.2 Overview of techniques

Here we provide a brief overview of the proof to compute an $\exp(-O(k \log n))$-approximate PML distribution. As discussed in the related work, both [CSS19a, ACSS20] analyzed the same convex program; [ACSS20] showed that this convex program approximates the probability of a profile for a fixed distribution up to a multiplicative factor of $\exp(-O(k \log n))$. However in the rounding step, their algorithms incurred a loss of $\exp(-O(n^{2/3} \log n))$ and $\exp(-O(\sqrt{n} \log n))$ respectively. Computing an improved $\exp(-O(k \log n))$-approximate PML distribution required a better rounding algorithm which in turn posed several challenges. We address these challenges by leveraging interesting sparsity structure in the convex relaxation of PML [ACSS20, CSS19a] (Lemma 4.3) and provide a novel matrix rounding algorithm (Theorem 4.4).

In our rounding algorithm, we first leverage homogeneity in the convex relaxation of PML and properties of basic feasible solutions of a linear program to efficiently obtain a sparse approximate solution to the convex relaxation. This reduces the problem of computing the desired approximate PML distribution to a particular matrix rounding problem where we need to "round down" a matrix of non-negative reals to another one with integral row and column sums without changing the entries too much ($O(k)$ overall) in $\ell_1$. Perhaps surprisingly, we show that this is always possible by reduction to a combinatorial problem which we solve by combining seemingly disparate theorems from combinatorics and graph theory. Further, we show that this rounding can be computed efficiently by employing algorithms for enumerating near-minimum-cuts of a graph [KS96].

## 3 Convex Relaxation to PML

Here we define the convex program that approximates the PML objective. This convex program was initially introduced in [CSS19a] and rigorously analyzed in [CSS19a, ACSS20]. We first describe the notation and later state the theorem in [ACSS20] that captures the guarantees of the convex program.

**Probability discretization:** Let $\mathbf{R} \overset{\text{def}}{=} \{\mathbf{r}_i\}_{i \in [1, \ell]}$ be a finite discretization of the probability space, where $\mathbf{r}_i = \frac{1}{2n^2}(1 + \alpha)^i$ for all $i \in [1, \ell - 1]$, $\mathbf{r}_\ell = 1$ and $\ell \overset{\text{def}}{=} |\mathbf{R}|$ be such that $\frac{1}{2n^2}(1 + \alpha)^\ell > 1$; therefore $\ell = O(\frac{\log n}{\alpha})$. Let $\mathbf{r} \in \mathbb{Z}_+^\ell$ be a vector where the $i$'th element is equal to $\mathbf{r}_i$. We call $\mathbf{q} \in [0, 1]_\mathbb{R}^{\mathcal{D}}$ a *pseudo-distribution* if $\|\mathbf{q}\|_1 \leq 1$ and a *discrete pseudo-distribution* with respect to $\mathbf{R}$ if all its entries are in $\mathbf{R}$ as well. We use $\Delta_{pseudo}^{\mathcal{D}}$ and $\Delta_{\mathbf{R}}^{\mathcal{D}}$ to denote the set of all pseudo-distributions and discrete pseudo-distributions with respect to $\mathbf{R}$ respectively. For all probability terms defined involving distributions $\mathbf{p}$, we extend those definitions to pseudo distributions $\mathbf{q}$ by replacing $\mathbf{p}_x$ with $\mathbf{q}_x$ everywhere. The effect of discretization is captured by the following lemma.

**Lemma 3.1** (Lemma 4.4 in [CSS19a]). *For any profile $\phi \in \Phi^n$ and distribution $\boldsymbol{p} \in \Delta^{\mathcal{D}}$, there exists $\boldsymbol{q} \in \Delta_{\boldsymbol{R}}^{\mathcal{D}}$ that satisfies $\mathbb{P}(\boldsymbol{p}, \phi) \geq \mathbb{P}(\boldsymbol{q}, \phi) \geq \exp(-\alpha n - 6) \mathbb{P}(\boldsymbol{p}, \phi)$ and therefore,*

$$\max_{\boldsymbol{p} \in \Delta^{\mathcal{D}}} \mathbb{P}(\boldsymbol{p}, \phi) \geq \max_{\boldsymbol{q} \in \Delta_{\boldsymbol{R}}^{\mathcal{D}}} \mathbb{P}(\boldsymbol{q}, \phi) \geq \exp(-\alpha n - 6) \max_{\boldsymbol{p} \in \Delta^{\mathcal{D}}} \mathbb{P}(\boldsymbol{p}, \phi).$$

For any probability discretization set $\mathbf{R}$, profile $\phi$ and $\mathbf{q} \in \Delta_{\mathbf{R}}^{\mathcal{D}}$, we define the following sets that help lower and upper bound the PML objective by a convex program.

$$\mathbf{Z}_{\mathbf{R}}^{\phi} \overset{\text{def}}{=} \left\{ \mathbf{S} \in \mathbb{R}_{\geq 0}^{\ell \times [0, k]} \,\Big|\, \mathbf{S1} \in \mathbb{Z}_+^\ell, [\mathbf{S}^\top \mathbf{1}]_j = \phi_j \text{ for all } j \in [1, k] \text{ and } \mathbf{r}^\top \mathbf{S1} \leq 1 \right\}, \tag{2}$$

$$\mathbf{Z}_{\mathbf{R}}^{\phi, frac} \overset{\text{def}}{=} \left\{ \mathbf{S} \in \mathbb{R}_{\geq 0}^{\ell \times [0, k]} \,\Big|\, [\mathbf{S}^\top \mathbf{1}]_j = \phi_j \text{ for all } j \in [1, k] \text{ and } \mathbf{r}^\top \mathbf{S1} \leq 1 \right\}, \tag{3}$$

where in the above definitions the 0'th column corresponds to domain elements with frequency 0 (unseen) and we use $\mathbf{m}_0 \overset{\text{def}}{=} 0$. We next define the objective of the convex program. Let $\mathbf{C}_{ij} \overset{\text{def}}{=} \mathbf{m}_j \log \mathbf{r}_i$ and for any $\mathbf{S} \in \mathbb{R}_{\geq 0}^{\ell \times [0, k]}$ define,

$$\mathbf{g}(\mathbf{S}) \overset{\text{def}}{=} \exp \left( \sum_{i \in [1, \ell], j \in [0, k]} [\mathbf{C}_{ij} \mathbf{X}_{ij} - \mathbf{X}_{ij} \log \mathbf{X}_{ij}] + \sum_{i \in [1, \ell]} [\mathbf{X1}]_i \log[\mathbf{X1}]_i \right). \tag{4}$$

The function $\mathbf{g}(\mathbf{S})$ approximates the $\mathbb{P}(\mathbf{q}, \phi)$ term and the following theorem summarizes this result.

**Theorem 3.2** (Theorem 6.7 and Lemma 6.9 in [ACSS20]). *Let $\mathbf{R}$ be a probability discretization set. Given a profile $\phi \in \Phi^n$ with $k$ distinct frequencies the following inequalities hold,*

$$\exp\left(-O(k \log n)\right) \cdot C_\phi \cdot \max_{\mathbf{S} \in \mathbf{Z}_{\mathbf{R}}^\phi} \boldsymbol{g}(\boldsymbol{S}) \leq \max_{\boldsymbol{q} \in \Delta_{\mathbf{R}}^{\mathcal{D}}} \mathbb{P}(\boldsymbol{q}, \phi) \leq \exp\left(O(k \log n)\right) \cdot C_\phi \cdot \max_{\mathbf{S} \in \mathbf{Z}_{\mathbf{R}}^\phi} \boldsymbol{g}(\boldsymbol{S}) , \quad (5)$$

$$\max_{\boldsymbol{q} \in \Delta_{\mathbf{R}}^{\mathcal{D}}} \mathbb{P}(\boldsymbol{q}, \phi) \leq \exp\left(O(k \log n)\right) \cdot C_\phi \cdot \max_{\mathbf{S} \in \mathbf{Z}_{\mathbf{R}}^{\phi, frac}} \boldsymbol{g}(\boldsymbol{S}) , \quad (6)$$

*where $C_\phi \overset{\text{def}}{=} \frac{n!}{\prod_{j \in [1,k]} (\boldsymbol{m}_j!)^{\phi_j}}$ is a term that only depends on the profile.*

See Appendix A.1 for citations related to convexity of the function $\mathbf{g}(\mathbf{S})$ and running time to solve the convex program. For any $\mathbf{S} \in \mathbf{Z}_{\mathbf{R}}^\phi$, define a pseudo-distributions associated with it as follows.

**Definition 3.3.** For any $\mathbf{S} \in \mathbf{Z}_{\mathbf{R}}^\phi$, the discrete pseudo-distribution $\mathbf{q_S}$ associated with $\mathbf{S}$ and $\mathbf{R}$ is defined as follows: For any arbitrary $\sum_{j \in [0,k]} \mathbf{S}_{i,j}$ number of domain elements assign probability $\mathbf{r}_i$.

Further $\mathbf{p_S} \overset{\text{def}}{=} \mathbf{q_S}/\|\mathbf{q_S}\|_1$ is the distribution associated with $\mathbf{S}$ and $\mathbf{R}$.

Note that $\mathbf{q_S}$ is a valid pseudo-distribution because of the third condition in Equation (2) and these pseudo distributions $\mathbf{p_S}$ and $\mathbf{q_S}$ satisfy the following lemma.

**Lemma 3.4** (Theorem 6.7 in [ACSS20]). *Let $\mathbf{R}$ and $\phi \in \Phi^n$ be any probability discretization set and a profile respectively. For any $\mathbf{S} \in \mathbf{Z}_{\mathbf{R}}^\phi$, the discrete pseudo distribution $\boldsymbol{q_S}$ and distribution $\boldsymbol{p_S}$ associated with $\mathbf{S}$ and $\mathbf{R}$ satisfies:* $\exp\left(-O(k \log n)\right) C_\phi \cdot \boldsymbol{g}(\boldsymbol{S}) \leq \mathbb{P}(\boldsymbol{q}, \phi) \leq \mathbb{P}(\boldsymbol{p}, \phi)$ .

## 4 Algorithm and Proof Sketch of Theorem 2.1

Here we provide the algorithm to compute an $\exp\left(-O(k \log n)\right)$-approximate PML distribution, where $k$ is the number of distinct frequencies. We use the convex relaxation from Section 3; the maximizer of this convex program is a matrix $\mathbf{S} \in \mathbf{Z}_{\mathbf{R}}^{\phi, frac}$ and its $i$'th row sum denotes the number of domain elements with probability $\mathbf{r}_i$. As the row sums are not necessarily integral, we wish to round $\mathbf{S}$ to a new matrix $\mathbf{S}'$ that has integral row sums and $\mathbf{S}' \in \mathbf{Z}_{\mathbf{R}'}^\phi$ for some probability discretization set $\mathbf{R}'$. Our algorithm does this rounding and incurs only a loss of $\exp\left(-O(k \log n)\right)$ in the objective; finally the distribution associated with $\mathbf{S}'$ and $\mathbf{R}'$ is the desired $\exp\left(-O(k \log n)\right)$-approximate PML. We first provide a general algorithm that holds for any probability discretization set $\mathbf{R}$ and the guarantees of this algorithm are stated below.

**Theorem 4.1.** *Given a profile $\phi \in \Phi^n$ with $k$ distinct observed frequencies and $\mathbf{R}$, there exists an algorithm that runs in polynomial of $n$ and $|\mathbf{R}|$ time and returns a distribution $\mathbf{p}'$ that satisfies,*

$$\mathbb{P}\left(\boldsymbol{p}', \phi\right) \geq \exp\left(-O(k \log n)\right) \max_{\boldsymbol{q} \in \Delta_{\mathbf{R}}^{\mathcal{D}}} \mathbb{P}\left(\boldsymbol{q}, \phi\right) .$$

For an appropriately chosen $\mathbf{R}$, the above theorem immediately proves Theorem 2.1 and we defer its proof to Appendix A.4. In the remainder of this section we focus our attention towards the proof of Theorem 4.1 and we next provide the algorithm that satisfies the guarantees of this theorem.

---

**Algorithm 1** ApproximatePML($\phi, \mathbf{R}$)

---

1: Solve $\mathbf{S}' = \arg\max_{\mathbf{S} \in \mathbf{Z}_{\mathbf{R}}^{\phi, frac}} \log \mathbf{g}(\mathbf{S})$.      ▷ Step 1
2: $\mathbf{S}'' = \text{Sparse}(\mathbf{S}')$.      ▷ Step 2
3: $(\mathbf{S}'', \mathbf{B}'') = \text{MatrixRound}(\mathbf{S}'')$.      ▷ Step 3
4: $(\mathbf{S}^{\text{ext}}, \mathbf{R}^{\text{ext}}) = \text{CreateNewProbabilityValues}(\mathbf{S}'', \mathbf{B}'', \mathbf{R})$.      ▷ Step 4
5: Return distribution $\mathbf{p}'$ with respect to $\mathbf{S}^{\text{ext}}$ and $\mathbf{R}^{\text{ext}}$ (See Definition 3.3).      ▷ Step 5

---

We divide the analysis of the above algorithm into 5 main steps. See Lemma 3.4 for the guarantees of Step 5 and here we state results for the remaining steps; we later combine it all to prove Theorem 4.1.

**Lemma 4.2** ([CSS19a, ACSS20]). *Step 1 of the algorithm can be implemented in $\widetilde{O}(|\boldsymbol{R}|\,k^2)$ time and the maximizer $\boldsymbol{S}'$ satisfies: $C_\phi \cdot \boldsymbol{g}(\boldsymbol{S}') \geq \exp\left(O\left(-k\log n\right)\right)\max_{\boldsymbol{q}\in\Delta_{\boldsymbol{R}}^{\mathcal{P}}}\mathbb{P}(\boldsymbol{q},\phi)$.*

The running time follows from Theorem 4.17 in [CSS19a] and the guarantee of the maximizer follows from Lemma 6.9 in [ACSS20]. The lemma statements for the remaining steps are written in a general setting; we later invoke each of these lemmas in the context of the algorithm to prove Theorem 4.1.

**Lemma 4.3** (Sparse solution). *For any $\boldsymbol{A}\in\boldsymbol{Z}_{\boldsymbol{R}}^{\phi,frac}$, the algorithm $\mathrm{Sparse}(\boldsymbol{A})$ runs in $\widetilde{O}(|\boldsymbol{R}|\,k^\omega)$ time and returns a solution $\boldsymbol{A}'\in\boldsymbol{Z}_{\boldsymbol{R}}^{\phi,frac}$ such that $\boldsymbol{g}(\boldsymbol{A}')\geq\boldsymbol{g}(\boldsymbol{A})$ and $\left|\{i\in[1,\ell]\mid[\boldsymbol{A}'\overrightarrow{1}]_i>0\}\right|\leq k+1$.*

We defer description of the algorithm $\mathrm{Sparse}(\boldsymbol{X})$ and the proof to Appendix A.1. In the proof, we use homogeneity of the convex program to write an LP whose optimal basic feasible solution satisfies the lemma conditions.

**Theorem 4.4.** *For a matrix $\boldsymbol{A}\in\mathbb{R}_{\geq 0}^{s\times t}$, the algorithm $\mathrm{MatrixRound}(\boldsymbol{A})$ runs in time polynomial in $s,t$ and returns a matrix $\boldsymbol{B}\in\mathbb{R}_{\geq 0}^{s\times t}$ such that $\boldsymbol{B}_{ij}\leq\boldsymbol{A}_{ij}\;\forall\,i\in[s],j\in[t]$, $\boldsymbol{B}\overrightarrow{1}\in\mathbb{Z}_+^s$, $\boldsymbol{B}^\top\overrightarrow{1}\in\mathbb{Z}_+^t$ and $\sum_{i,j}(\boldsymbol{A}_{ij}-\boldsymbol{B}_{ij})\leq O(s'+t')$, where $s',t'$ denote the number of non-zeros rows and columns.*

For continuity of reading, we defer the description of $\mathrm{MatrixRound}(\mathbf{A})$ and its proof to Section 4.1.

**Lemma 4.5** (Lemma 6.13 in [ACSS20]). *For any $\boldsymbol{A}\in\boldsymbol{Z}_{\boldsymbol{R}}^{\phi,frac}\subseteq\mathbb{R}_{\geq 0}^{\ell\times[0,k]}$ and $\boldsymbol{B}\in\mathbb{R}_{\geq 0}^{\ell\times[0,k]}$ such that $\boldsymbol{B}_{ij}\leq\boldsymbol{A}_{ij}$ for all $i\in[\ell],j\in[0,k]$, $\boldsymbol{B}\overrightarrow{1}\in\mathbb{Z}_+^\ell$, $\boldsymbol{B}^\top\overrightarrow{1}\in\mathbb{Z}_+^{[0,k]}$ and $\sum_{i\in[\ell],j\in[0,k]}(\boldsymbol{A}_{ij}-\boldsymbol{B}_{ij})\leq t$. The algorithm $\mathrm{CreateNewProbabilityValues}(\boldsymbol{A},\boldsymbol{B},\boldsymbol{R})$ runs in polynomial time and returns a solution $\boldsymbol{A}'$ and a probability discretization set $\boldsymbol{R}'$ such that $\boldsymbol{A}'\in\boldsymbol{Z}_{\boldsymbol{R}'}^\phi$ and $\boldsymbol{g}(\boldsymbol{A}')\geq\exp\left(-O\left(t\log n\right)\right)\boldsymbol{g}(\boldsymbol{A})$ .*

The algorithm $\mathrm{CreateNewProbabilityValues}$ is the same algorithm from [ACSS20] and the above lemma is a simplified version of Lemma 6.13 in [ACSS20]; see Appendix A.3 for its proof.

The proof of Theorem 4.1 follows by combining results for each step and we defer it to Appendix A.4.

## 4.1 Matrix rounding algorithm and proof sketch of Theorem 4.4

In this section we prove Theorem 4.4. Given a matrix $\mathbf{A}\in\mathbb{R}_{\geq 0}^{s\times t}$, our goal is to produce a rounded-down matrix $\mathbf{B}$ with integer row and column sums, such that $0\leq\mathbf{B}\leq\mathbf{A}$ (entry wise) and the total amount of rounding $\sum_{i,j}(\mathbf{A}_{ij}-\mathbf{B}_{ij})$ is bounded by $O(s'+t')$, where $s',t'$ are the number of nonzero rows and columns respectively. For simplicity we may assume $s=s'$ and $t=t'$ by simply dropping the zero rows and columns from $\mathbf{A}$ and re-appending them to the resulting $\mathbf{B}$. As our first step, we reduce the problem to a statement about graphs. Below we use $\deg_F(v)$ to denote the number of edges adjacent to a vertex $v$ within a set of edges $F$.

**Lemma 4.6.** *Suppose that $G=(V,E)$ is a bipartite graph and $k$ is a positive integer. There exists a polynomial time algorithm that outputs a subgraph $F\subseteq E$, such that $\deg_F(v)=0$ modulo $k$ for every vertex $v$, and $|E-F|\leq O(k|V|)$.*

*Proof of Lemma 4.6 $\implies$ Theorem 4.4.* Let $k=\min(s,t)$. Given $\mathbf{A}$ we produce a bipartite graph with $s$ and $t$ vertices on two sides; for every entry $\mathbf{A}_{ij}$ we round down to the nearest integer multiple of $1/k$, say $c_{ij}/k$, and introduce $c_{ij}$ parallel edges between vertices $i$ and $j$ of the bipartite graph. Now Lemma 4.6 produces a subgraph $F$, and we let $\mathbf{B}_{ij}$ be $1/k$ times the number of edges left in $F$ between $i,j$. By Lemma 4.6, $\mathbf{B}$ will have integer row and column sums, and $0\leq\mathbf{B}\leq\mathbf{A}$. We next show that the total amount of rounding is bounded by $O(s+t)$.

Notice that when rounding each entry of $\mathbf{A}$ down to $c_{ij}/k$, the total amount of change is at most $st/k=O(s+t)$. By the guarantee that $|E-F|\leq O(k|V|)$, the total amount of rounding in the second step is also bounded by $O(k(s+t))/k=O(s+t)$. $\qquad\square$

So it remains to prove Lemma 4.6. As our main tool, we will use a result from [Tho14] which was obtained by reduction to an earlier result from [LTWZ13]. Roughly, this result says that as long as $G$ is sufficiently connected, we can choose a subgraph whose degrees are *arbitrary* values modulo $k$.

**Lemma 4.7** ([Tho14, Theorem 1]). *Suppose that $G=(V,E)$ is a bipartite $(3k-3)$-edge-connected graph. Suppose that $f:V\to\{0,\ldots,k-1\}$ is an arbitrary function, with the restriction that the*

*sum of $f$ on either side of the bipartite graph $G$ yields the same result modulo $k$. Then, there is a subgraph $F \subseteq E$, such that for each vertex $v$, $\deg_F(v) = f(v)$ modulo $k$.*

Note that $(3k-3)$-edge-connectivity means that for every cut, i.e., every partitioning of vertices into two nonempty sets $S, S^c$, the number of edges between $S$ and $S^c$ is $\geq 3k - 3$. We show that Lemma 4.7 can also be made constructive, giving the polynomial time guarantee for Lemma 4.6.

**Lemma 4.8.** *There is a polynomial time algorithm that produces the subgraph of Lemma 4.7.*

We defer the proof of Lemma 4.8 to Appendix A.2. At a high level, the proof of Lemma 4.7 works by formulating an assumption about the graph that is more general and more nuanced than edge-connectivity; instead of a constant lower bound on every cut, this assumption puts a cut-specific lower bound on each cut, the details of which can be found in Appendix A.2. The rest of the argument follows a clever induction. To make this argument constructive, we show how to check the nuanced variant of edge-connectivity in polynomial time. We do this by proving that only cuts of size smaller than a constant multiple of the minimum cut have to checked, and these can be enumerated in polynomial time [KS96].

Note that Lemma 4.7 does not guarantee anything about $|E - F|$, even when $f$ is the zero function (the empty subgraph is actually a valid answer in that case). We will fix this using a theorem of [NW61]. We will first prove Lemma 4.6 with the extra assumption that $G$ is $6k$-edge-connected, and then prove the general case.

*Proof of Lemma 4.6 when $G$ is $6k$-edge-connected.* By a famous theorem due to [NW61], a $6k$-edge-connected graph contains $6k/2 = 3k$ edge-disjoint spanning trees. Moreover the union of these $3k$ edge-disjoint spanning trees can be found in polynomial time by matroid partitioning algorithms [GW92]. Let $H$ be the subgraph formed by these $3k$ edge-disjoint spanning trees. We will ensure that all edges outside $H$ are included in $F$; as a consequence, we will automatically get that $|E - F|$ is bounded by the number of edges in $H$, which is at most $3k(|V| - 1) = O(k|V|)$.

Let $H^c$ denote the complement of $H$ in $G$. Define the function $f : V \to \{0, \ldots, k-1\}$ in the following way: let $f(v)$ be $-\deg_{H^c}(v)$ modulo $k$. Note that $f$ has the same sum on either side of the bipartite graph, modulo $k$. We will apply Lemmas 4.7 and 4.8 to the graph $H$ (which is $3k \geq (3k - 3)$-edge-connected) and the function $f$. Then we take the union of the subgraph returned by Lemma 4.8 and $H^c$ and output the result as $F$. Then $\deg_F(v) = \deg_{H^c}(v) + f(v) = 0$ modulo $k$, for every vertex $v$. Note again that since we only deleted edges in $H$ to get $F$, the total number of edges we have removed can be at most $O(k|V|)$. □

We have shown Lemma 4.6 for highly-connected graphs and the proof for the general case follows by partitioning the graph into union of vertex-disjoint highly-connected subgraphs while removing a small number of edges. We defer the proof for this general case to Appendix A.2.

# 5 Algorithm, Proof Sketch of Theorem 2.4 and Experiments

Here we present a simpler rounding algorithm that further provides a faster implementation of the pseudo PML approach with provable guarantees. Similar to Section 4, we first provide an algorithm with respect to a probability discretization set $\mathbf{R}$ that proves Theorem 5.1; we later choose the discretization set carefully to prove Theorem 2.4. We perform experiments in Section 5.1 to analyze the performance of this rounding algorithm empirically. We defer all remaining details to Appendix B.

**Theorem 5.1.** *Given a probability discretization set $\mathbf{R}$ ($\ell \overset{\text{def}}{=} |\mathbf{R}|$) and a profile $\phi \in \Phi^n$ with $k$ distinct frequencies, there is an algorithm that runs in time $\widetilde{O}(\ell k^\omega)$ and returns a distribution $\mathbf{p}'$ such that,*

$$\mathbb{P}(\mathbf{p}', \phi) \geq \exp\left(-O((\mathbf{r}_{\max} - \mathbf{r}_{min})n + k \log(\ell n))\right) \max_{\mathbf{q} \in \Delta_{\mathbf{R}}^{\mathcal{D}}} \mathbb{P}(\mathbf{q}, \phi) .$$

For an appropriately chosen $\mathbf{R}$, the above theorem immediately proves Theorem 2.4 and we defer both their proofs to Appendix B.1. We now present the algorithm that proves Theorem 5.1.

**Algorithm 2** ApproximatePML2($\phi, \mathbf{R}$)

---

1: Solve $\mathbf{X} = \arg\max_{\mathbf{S} \in \mathbf{Z}_{\mathbf{R}}^{\phi, frac}} \log \mathbf{g}(\mathbf{S})$ and let $\mathbf{X}' = \mathrm{Sparse}(\mathbf{X})$.  ▷ Step 1

2: Let $\mathbf{S}'$ be the sub matrix of $\mathbf{X}'$ corresponding to its non-zero rows.  ▷ Step 2

3: Let $\mathbf{R}'$ denote the elements in $\mathbf{R}$ corresponding to non-zero rows of $\mathbf{X}'$. Let $\ell' \overset{\text{def}}{=} |\mathbf{R}'|$.  ▷ Step 3

4: **for** $i = 1 \ldots \ell' - 1$ **do**  ▷ Step 4

5: $\quad \mathbf{S}_{i,j}^{\mathrm{ext}} = \mathbf{S}_{i,j}' \frac{\lfloor \|\mathbf{S}_i'\|_1 \rfloor}{\|\mathbf{S}_i'\|_1}$ for all $j \in [0, k]$.  ▷ Step 5

6: $\quad \mathbf{S}_{i+1,j}' = \mathbf{S}_{i+1,j}' + (\mathbf{S}_{i,j}' - \mathbf{S}_{i,j}^{\mathrm{ext}})$ for all $j \in [0, k]$.  ▷ Step 6

7: **end for**  ▷ Step 7

8: $\mathbf{S}_{\ell',j}^{\mathrm{ext}} = \mathbf{S}_{\ell',j}' \frac{\lfloor \|\mathbf{S}_{\ell'}'\|_1 \rfloor}{\|\mathbf{S}_{\ell'}'\|_1}$ for all $j \in [0, k]$.  ▷ Step 8

9: Let $c = \sum_{i \in [1, \ell']} \mathbf{r}_i' \|\mathbf{S}_i^{\mathrm{ext}}\|_1$, where $\mathbf{r}_i'$ are the elements of $\mathbf{R}'$.  ▷ Step 9

10: Define $\mathbf{R}^{\mathrm{ext}} = \{\mathbf{r}_i''\}_{i \in [1, \ell']}$, where $\mathbf{r}_i'' = \frac{\mathbf{r}_i'}{c}$ for all $i \in [1, \ell']$.  ▷ Step 10

11: Return distribution $\mathbf{p}'$ with respect to $\mathbf{S}^{\mathrm{ext}}$ and $\mathbf{R}^{\mathrm{ext}}$ (See Definition 3.3).  ▷ Step 11

---

## 5.1 Experiments

Here we present experimental results for entropy estimation. We analyze the performance of the PseudoPML approach implemented using our rounding algorithm with the other state-of-the-art estimators. Each plot depicts the performance of various algorithms for estimating entropy of different distributions with domain size $N = 10^5$. The x-axis corresponds to the sample size (in logarithmic scale) and the y-axis denotes the root mean square error (RMSE). Each data point represents 50 random trials. "Mix 2 Uniforms" is a mixture of two uniform distributions, with half the probability mass on the first $N/10$ symbols and the remaining mass on the last $9N/10$ symbols, and $\mathrm{Zipf}(\alpha) \sim 1/i^\alpha$ with $i \in [N]$. MLE is the naive approach of using empirical distribution with correction bias; all the remaining algorithms are denoted using bibliographic citations.

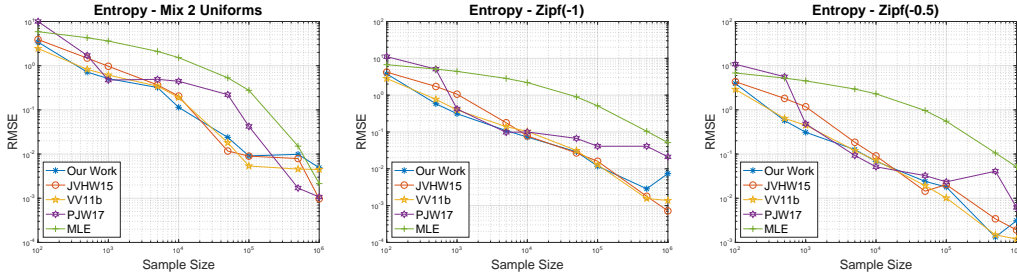

Figure 1: Experimental results for entropy estimation.

In the above experiment, note that the error achieved by our estimator is competitive with the other state-of-the-art estimators. As for the running times in practice, the other approaches tend to perform better than the current implementation of our algorithm. To further improve the running time of our approach or any other provable PML based approaches involves building an efficient practical solver for the convex optimization problem [CSS19a, ACSS20] stated in the first step[5] of our Algorithm 1; we think building such an efficient practical solver is an important research direction.

In Appendix B.4, we provide experiments for other distributions, compare the performance of the PseudoPML approach implemented using our algorithm with a heuristic approximate PML algorithm [PJW17] and provide all the implementation details.

## Broader Impact

Symmetric property estimation has a broad range of applications, ranging from ecology [Cha84, CL92, BF93, CCG$^+$12], to physics [VBB$^+$12], to neuroscience [RWdRvSB99], and beyond [HJWW17, HJM17, AOST14, RVZ17, ZVV$^+$16, WY16b, RRSS07, WY15, OSW16, VV11b, WY16a, JVHW15, JHW16, VV11a]. By providing new, broadly applicable, computationally efficient tools for obtaining higher accuracy estimates to symmetric properties this work could enable a variety of applications in machine learning and the sciences more broadly. Though the primary contributions of this work are theoretical, the preliminary experimental results show that this work could ultimately lead to obtaining higher quality answers to statistical questions at lower computational cost, with less manual tuning to the particular statistical question of interest. This could ultimately help save time, energy, and the many costs associated with data collection. There are always risks in widespread application of statistical tools, we are unaware of any particular biases or harm from the methods proposed. Further research may be required before the results of this paper can have a broad societal impact.

## Acknowledgments

We thank Alon Orlitsky and Yi Hao for helpful clarifications and discussions.

## Sources of Funding

Researchers on this project were supported by a Microsoft Research Faculty Fellowship, NSF CAREER Award CCF-1844855, NSF Grant CCF-1955039, a Simons Investigator Award, a Google Faculty Research Award, an Amazon Research Award, a PayPal research gift, a Sloan Research Fellowship, a Stanford Data Science Scholarship and a Dantzig-Lieberman Operations Research Fellowship.

## Competing Interests

The authors declare no competing interests.

## Footnotes

[1] Sample optimality is up to constant factors. See [ADOS16] for details.

[2] We use $\epsilon \gg n^{-c}$ to denote $\epsilon > n^{-c+\alpha}$ for any constant $\alpha > 0$.

[3]Theorem 3 in [HO20] discuss instead exact PML and the authors discuss the approximate PML case in the comments; we confirmed the sufficiency of approximate PML claimed in the theorem through private communication with the authors.

[4]Please refer [HO20] for general success probability $1 - \delta$; our work also holds for the general case.

[5]In our current implementation, we use CVX[GB14] with package CVXQUAD[FSP17] to solve the convex program stated in the first step of Algorithm 1.

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
