[Supplementary Material]

# A Remaining Proofs from Section 4

Here we provide proofs for all the results in Section 4 that were excluded in the main paper. For each of these results we dedicate a subsection that provides further details. Combining all these results from different subsections, in Appendix A.4 we provide the proof for our main result (Theorem 2.1).

## A.1 Properties of Convex Program and Proof of Lemma 4.3

Here we prove important properties of our convex program. For convenience, we define the negative log of function $\mathbf{g}(\mathbf{X})$,

$$\mathbf{f}(\mathbf{X}) \stackrel{\text{def}}{=} \sum_{i \in [1,\ell], j \in [0,k]} [-\mathbf{C}_{ij}\mathbf{X}_{ij} + \mathbf{X}_{ij}\log\mathbf{X}_{ij}] - \sum_{i \in [1,\ell]} [\mathbf{X1}]_i \log[\mathbf{X1}]_i = -\log\mathbf{g}(\mathbf{X}) . \quad (7)$$

In the remainder we prove and state interesting properties of this function that helps us construct sparse approximate solutions. We start by recalling properties showed in [CSS19a].

**Lemma A.1** (Lemma 4.16 in [CSS19a]). *Function $\mathbf{f}(\mathbf{X})$ is convex in $\mathbf{X}$.*

**Theorem A.2** (Theorem 4.17 in [CSS19a]). *Given a profile $\phi \in \Phi^n$ with $k$ distinct frequencies, the optimization problem $\min_{\mathbf{X} \in \mathbf{Z}_R^{\phi, frac}} \mathbf{f}(\mathbf{X})$ can be solved in time $\widetilde{O}(k^2|\mathbf{R}|)$.*

The function $\mathbf{f}(\mathbf{X})$ is separable in each row and we define following notation to capture it.

$$\mathbf{f}_i(\mathbf{X}_i) \stackrel{\text{def}}{=} \sum_{j \in [0,k]} [-\mathbf{C}_{ij}\mathbf{X}_{ij} + \mathbf{X}_{ij}\log\mathbf{X}_{ij}] - [\mathbf{X1}]_i \log([\mathbf{X1}]_i) \quad \text{and} \quad \mathbf{f}(\mathbf{X}) = \sum_{i \in [1,\ell]} \mathbf{f}_i(\mathbf{X}_i) .$$

The function $\mathbf{f}_i(\mathbf{X}_i)$ defined above is 1-homogeneous and is formally shown next.

**Lemma A.3.** *For any fixed vector $c \in \mathbb{R}^{[0,k]}$, the function $\mathbf{h}(v) = \sum_{j \in [0,k]} [c_j v_j + v_j \log v_j] - v^\top \overrightarrow{1} \log v^\top \overrightarrow{1}$ is 1-homogeneous, that is, $\mathbf{h}(\alpha \cdot v) = \alpha \cdot \mathbf{h}(v)$ for all $v \in \mathbb{R}_{\geq 0}^{[0,k]}$ and $\alpha \in \mathbb{R}_{\geq 0}$.*

*Proof.* Consider any vector $v \in \mathbb{R}_{\geq 0}^{k+1}$ and scalar $\alpha \in \mathbb{R}_{\geq 0}$ we have,

$$\mathbf{h}(\alpha \cdot v) = \sum_{j \in [0,k]} [c_j(\alpha v_j) + (\alpha v_j)\log(\alpha v_j)] - (\alpha v)^\top \overrightarrow{1} \log(\alpha v)^\top \overrightarrow{1},$$

$$= \sum_{j \in [0,k]} [c_j(\alpha v_j) + \alpha v_j \log v_j + \alpha v_j \log\alpha] - (\alpha v)^\top \overrightarrow{1} \log v^\top \overrightarrow{1} - (\alpha v)^\top \overrightarrow{1} \log\alpha,$$

$$= \sum_{j \in [0,k]} [c_j(\alpha v_j) + \alpha v_j \log v_j] - \alpha v^\top \overrightarrow{1} \log v^\top \overrightarrow{1} = \alpha \cdot \mathbf{h}(v) .$$

The above derivation satisfies the conditions of the lemma and we conclude the proof. □

In the remainder of this section, we provide the proof of Lemma 4.3 and the description of the algorithm Sparse is included inside the proof. The Lemma 4.3 in the notation of $\mathbf{f}(\cdot)$ can be equivalently written as follows.

**Lemma A.4** (Lemma 4.3). *For any $\mathbf{X} \in \mathbf{Z}_R^{\phi, frac}$, the algorithm $\text{Sparse}(\mathbf{X})$ runs in $\widetilde{O}(|\mathbf{R}| k^\omega)$ time and returns a solution $\mathbf{X}' \in \mathbf{Z}_R^{\phi, frac}$ such that $\mathbf{f}(\mathbf{X}') \leq \mathbf{f}(\mathbf{X})$ and $\left|\{i \in [1,\ell] \mid [\mathbf{X}'\overrightarrow{1}]_i > 0\}\right| \leq k + 1$.*

*Proof.* Let $\ell \stackrel{\text{def}}{=} |\mathbf{R}|$ and fix $\mathbf{X} \in \mathbf{Z}_R^{\phi, frac}$, consider the following solution $\mathbf{X}'_i = \alpha_i \mathbf{X}_i$ for all $i \in [1,\ell]$, where $\alpha \in \mathbb{R}_{\geq 0}^{[1,\ell]}$ and $\mathbf{X}_i, \mathbf{X}'_i$ denote the vectors corresponding to the $i$'th row of matrices $\mathbf{X}, \mathbf{X}'$ respectively. By Lemma A.3, each function $\mathbf{f}_i(\mathbf{X}_i)$ is 1-homogeneous and we get,

$$\mathbf{f}(\mathbf{X}') = \sum_{i \in [1,\ell]} \mathbf{f}_i(\mathbf{X}'_i) = \sum_{i \in [1,\ell]} \mathbf{f}_i(\alpha_i \mathbf{X}_i) = \sum_{i \in [1,\ell]} \alpha_i \mathbf{f}_i(\mathbf{X}_i) .$$

Let $\alpha \in \mathbb{R}_{\geq 0}^{[1,\ell]}$ be such that the following conditions hold,

$$\sum_{i \in [1,\ell]} \alpha_i \mathbf{X}_{i,j} = \phi_j \text{ for all } j \in [1,k] \text{ and } \sum_{i \in [1,\ell]} \alpha_i \mathbf{r}_i [\mathbf{X}\mathbf{1}]_i \leq 1 \ . \tag{8}$$

For the above set of equations, the solution $\alpha = \mathbf{1}$ is feasible as $\mathbf{X} \in \mathbf{Z}_{\mathbf{R}}^{\phi,frac}$. Further for any $\alpha$ satisfying the above inequalities, the corresponding matrix $\mathbf{X}'$ satisfies,

$$\sum_{i \in [1,\ell]} \mathbf{X}'_{i,j} = \sum_{i \in [1,\ell]} \alpha_i \mathbf{X}_{i,j} = \phi_j \text{ for all } j \in [1,k] \text{ and } \sum_{i \in [1,\ell]} \mathbf{r}_i [\mathbf{X}'\mathbf{1}]_i = \sum_{i \in [1,\ell]} \alpha_i \mathbf{r}_i [\mathbf{X}\mathbf{1}]_i \leq 1 \ .$$

Therefore $\mathbf{X}' \in \mathbf{Z}_{\mathbf{R}}^{\phi,frac}$ for all $\alpha \in \mathbb{R}_{\geq 0}^{[1,\ell]}$ that satisfy Equation (8). In the remainder of the proof we find a sparse $\alpha$ that satisfies the conditions of the lemma.

Consider the following linear program.

$$\min \alpha \in \mathbb{R}_{\geq 0}^{[1,\ell]} \sum_{i \in [1,\ell]} \alpha_i \mathbf{f}_i(\mathbf{X}_i) \ .$$

$$\text{such that,} \sum_{i \in [1,\ell]} \alpha_i \mathbf{X}_{i,j} = \phi_j \text{ for all } j \in [1,k] \text{ and } \sum_{i \in [1,\ell]} \alpha_i \mathbf{r}_i [\mathbf{X}\mathbf{1}]_i \leq 1 \ .$$

Note in the above optimization problem we fix $\mathbf{X} \in \mathbf{Z}_{\mathbf{R}}^{\phi,frac}$ and optimize over $\alpha$. Any basic feasible solution (BFS) $\alpha^*$ to the above LP, satisfies $|\{i \in [1,\ell] \mid \alpha_i^* > 0\}| \leq k+1$ as there are at most $k+1$ non-trivial constraints. Suppose we find a basic feasible solution $\alpha^*$ such that the corresponding matrix $\mathbf{X}'_i = \alpha_i^* \mathbf{X}_i$ for all $i \in [1,\ell]$ satisfies $\mathbf{f}(\mathbf{X}') \leq \mathbf{f}(\mathbf{X})$, then such a matrix $\mathbf{X}'$ is the desired solution that satisfies the conditions of the lemma. Therefore in the remainder of the proof, we discuss the running time to find such a BFS given a feasible solution to the LP.

Leveraging these insights, we design the following iterative algorithm. In each iteration $i$ we maintain a set $S_i \subseteq \mathbb{R}^{k+1}$ of $1 \leq k_i \leq k+1$ linearly independent rows of matrix $\mathbf{X}$. We update the solution $\alpha$ and try to set a non-zero coordinate of it to value zero while not increasing the objective. Our algorithm starts with $k_i = 1$ and $S_i$ to be the set containing an arbitrary row of $\mathbf{X}$ in iteration $i = 1$. The next iteration is computed by considering an arbitrary row $r$ of matrix $\mathbf{X}$ that corresponds to a non-zero coordinate in $\alpha$. Letting $\mathbf{A}_i \in \mathbb{R}^{(k+1) \times k_i}$ be the matrix where the columns are the vectors in $S_i$ we then consider the linear system $\mathbf{A}_i^\top \mathbf{A}_i x = r$. Whether or not there is such a solution can be computed in $O(k^\omega)$, where $\omega < 2.373$ is the matrix multiplication constant [Wil12, LG14, AV20] using fast matrix multiplication as in this time we can form the $(k+1) \times (k+1)$ matrix $\mathbf{A}_i^\top \mathbf{A}_i$ directly and then invert it. If this system has no solution we let $S_{i+1} = S_i \cup r$ and proceed to the next iteration as the lack of a solution proves that $S_i \cup r$ are linearly independent (as $S_i$ is linearly independent). Otherwise, we consider the vector $\alpha'$ in the null space of the transpose of $\mathbf{X}$ formed by setting $\alpha'_i$ to the value of $x_j$ for the associated rows and setting $\alpha'_i$ for the row corresponding to row $r$ to be $-1$. As $x$ is a solution to $\mathbf{A}_i^\top \mathbf{A}_i x = r$, clearly $\mathbf{X}^\top \alpha' = 0$. Now consider the solution $\alpha + c\alpha'$ for some scaling $c$. Since the objective and constraints are linear, there exists a direction, that is, sign of $c$ such that the objective is non-increasing and the solution $\alpha + c\alpha'$ satisfies all the constraints (Equation (8)). We start with $c = 0$ and keep increasing it in the direction where the objective in non-increasing till one of the following two conditions hold: either a new coordinate in the solution $\alpha + c\alpha'$ becomes zero or the objective value of the LP is infinity. In the first case, we update our current solution $\alpha$ to $\alpha + c\alpha'$ and repeat the procedure. As the goal our algorithm is to find a sparse solution, we fix the co-ordinates in $\alpha$ that have value zero and never change (or consider) them in the later iterations of our algorithm. We repeat this procedure till all the non-zero co-ordinates in $\alpha$ are considered at least once and the solution $\alpha$ returned at the end corresponds to a BFS that satisfies the desired conditions. As the total number of rows is at most $\ell$, our algorithm has at most $\ell$ iterations and each iteration takes only $O(k^\omega)$ time (note that we only update $\widetilde{O}(k)$ coordinates in each iteration). Therefore the final running time of the algorithm $\text{Sparse}$ is $\widetilde{O}(\ell k^\omega)$ time and we conclude the proof.

$\square$

## A.2 Remaining Parts of the Proof for Theorem 4.4

We first finish the proof of Lemma 4.6. That only leaves us with proving Lemma 4.8.

*Proof of Lemma 4.6 in the general case.* Since the input graph is arbitrary, we have no guarantee about edge-connectivity. We will show that we can remove $O(k|V|)$ edges from $G$ so that the remaining subgraph is a vertex-disjoint union of $6k$-edge-connected induced subgraphs. To do this, look at the connected components of $G$. Either they are all $6k$-edge-connected or at least one of them has a cut with $< 6k$ edges. Moreover we can check this in polynomial time (and find violating cuts if there are any) by a global minimum cut algorithm [Kar00]. If a component is not $6k$-edge-connected, remove all edges of the small cut, and repeat. Every time we remove the edges of a cut, the number of connected components increases by 1, so this can go on for at most $O(|V|)$ iterations. In each iteration, at most $6k$ edges are removed, so the total number of removed edges is $O(k|V|)$.

So by removing $O(k|V|)$ edges, we have transformed $G$ into a vertex-disjoint union of $6k$-edge-connected graphs. We simply apply the already-proved case of Lemma 4.6 to each of these components to get our desired result for the original graph $G$. □

In the remainder of this section we prove Lemma 4.8. We do this by showing how to make the proof of Lemma 4.7 due to [Tho14] algorithmic. [Tho14] reduced Lemma 4.7 to an earlier result by [LTWZ13] which we state below.

**Lemma A.5** ([LTWZ13, Theorem 1.12]). *Let $k \geq 3$ be an odd integer and $G = (V, E)$ a $(3k-3)$-edge connected undirected graph. For any given $\beta : V \to \{0, \ldots, k-1\}$ where $\sum_v \beta(v) \equiv 0$ (mod $k$), there is an orientation of $G$ which makes $\deg_{\text{out}}(v) - \deg_{\text{in}}(v)$ equal to $\beta(v)$ modulo $k$ for every vertex $v$.*

Here an orientation is an assignment of one of the two possible directions to each edge, and $\deg_{\text{out}}$ and $\deg_{\text{in}}$ count outgoing and incoming edges of a vertex in such an orientation. We simply note that the reduction of Lemma 4.7 to Lemma A.5, as stated in [Tho14], is already efficient. This is done by a simple transformation on $f$ from Lemma 4.7 to get $\beta$, and at the end a subgraph is extracted from an orientation by considering edges oriented from one side to the other. Since the reduction is efficient, we simply need to prove Lemma A.5 can be made efficient.

**Lemma A.6.** *There is a polynomial time algorithm that outputs the orientation of Lemma A.5.*

To obtain this algorithm, our strategy is to make the steps of the proof presented in [LTWZ13] (efficiently) constructive. [LTWZ13] prove Lemma A.5 by generalizing the statement and using a clever induction. To state this generalization, we need a definition from [LTWZ13].

**Definition A.7** ([LTWZ13]). Suppose that $k$ is an odd integer, and $G = (V, E)$ is an undirected graph. For a given function $\beta : V \to \{0, \ldots, k-1\}$, we define a set function $\tau : 2^V \to \{0, \pm 1, \ldots, \pm k\}$ by the following congruences

$$\tau(S) \equiv \sum_{v \in S} \beta(S) \pmod{k}$$

$$\tau(S) \equiv \sum_{v \in S} \deg(S) \pmod{2}$$

The two given congruences uniquely determine $\tau(S)$ modulo $2k$; this in turn is a unique element of $\{0, \pm 1, \ldots, \pm k\}$, except for $k$ and $-k$ which are the same value modulo $2k$. The choice of which value to take in this case is largely irrelevant, as we will mostly be dealing with $|\tau(\cdot)|$. Note that $\tau(S)$ is the same, modulo $2k$, as the number of edges going from $S$ to $S^c$ minus the number of edges going from $S^c$ to $S$ in any valid orientation as promised by Lemma A.5.

The definition of $\tau$ is used to give a generalization of Lemma A.5 that is proved by induction.

**Lemma A.8** ([LTWZ13, Theorem 3.1]). *Let $k$ be an odd integer, $G = (V, E)$ an undirected graph on at least 3 vertices, and $\beta : V \to \{0, \ldots, k-1\}$ be such that $\sum_v \beta(v) \equiv 0 \pmod{k}$. Let $z_0$ be a "special" vertex of $G$ whose adjacent edges are already pre-oriented in a specified way. Assume that $\tau$ is defined as in Definition A.7 and $V_0 = \{v \in V - \{z_0\} \mid \tau(\{v\}) = 0\}$; let $v_0$ be a vertex of minimum degree in $V_0$. If the following conditions are satisfied, then there is an orientation of edges, matching the pre-orientation of $z_0$, for which $\deg_{\text{out}}(v) - \deg_{\text{in}}(v) \equiv \beta(v) \pmod{k}$ for every $v$.*

    *1. $\deg(z_0) \leq (2k-2) + |\tau(\{z_0\})|,$*

2. $|E(S, S^c)| \geq (2k - 2) + |\tau(S)|$ *for every set* $S$ *where* $z_0 \notin S$, *and* $S \neq \emptyset, \{v_0\}, V - \{z_0\}$.

Here $E(S, S^c)$ is the set of edges between $S$ and $S^c$. Note that we always have $|\tau(\cdot)| \leq k$. So a $(3k - 3)$-edge-connected graph automatically satisfies condition 2 in Lemma A.8. Lemma 4.7 is proved by adding an isolated vertex $z_0$ and setting $\beta(z_0) = 0$, for which condition 1 is automatically satisfied.

The reason behind this generalization is the ability to prove it by induction. The authors of [LTWZ13] state this induction in the form of proof by contradiction. They consider a minimal counterexample, and argue the existence of a smaller counterexample. We do not state all of their proof again here, but note that all processes used to produce smaller counterexamples are readily efficiently implementable, except for one. In the proof of Theorem 3.1 in [LTWZ13], in Claim 1, the authors argue that for non-singleton $S$ the inequality in condition 2 of Lemma A.8 cannot be strict, or else the size of the problem can be reduced. They formally prove that a smallest counterexample must satisfy for $|S| \geq 2$,

$$|E(S, S^c)| \geq 2k + |\tau(S)| > (2k - 2) + |\tau(S)|. \tag{9}$$

In case a non-singleton does not satisfy the above inequality, the authors produce two smaller instances, once by contracting $S$ into a single vertex, and once by contracting $S^c$, and combining the resulting orientations together for all of $G$. The main barrier in making this into an efficient algorithm is *finding* the set $S$ that violates the inequality. A priori, it might seem like an exhaustive search over all subsets $S$ is needed, but we show that this is not the case.

We now show how to make this part algorithmic.

**Lemma A.9.** *Suppose that the graph* $G$ *satisfies the conditions of Lemma A.8. Then there is a polynomial time algorithm which produces a list of sets* $S_1, \ldots, S_m$ *for a polynomially bounded* $m$, *such that any violation of Eq.* (9) *must happen for some* $S_i$.

*Proof.* Our high-level strategy is to use the fact that condition 2 of Lemma A.8 implies $G$ is already sufficiently edge-connected. If $z_0, v_0$ did not exist, condition 2 would imply that $G$ is $(2k - 2)$-edge-connected. On the other hand any violation of Eq. (9) can only happen when $|E(S, S^c)| < 2k + k = 3k$. So it would be enough to simply produce a list of all near-minimum-cuts $S$ with $|E(S, S^c)| < 3k$. If $G$ was $(2k - 2)$-edge-connected, we could appeal to results of [KS96], who proved that for any constant $\alpha$, the number of cuts of size at most $\alpha$ times the minimum cut is polynomially bounded and all of them can be efficiently enumerated.

The one caveat is the existence of $v_0, z_0$, which might make $G$ not $(2k - 2)$-edge-connected. Note that the only cuts that can potentially be "small" are the singletons $\{v_0\}, \{z_0\}$. We can solve this problem by contracting the graph. We enumerate over the edges $e_1, e_2$ that are adjacent to $v_0, z_0$, and for every choice of $e_1, e_2$, we produce a new graph by contracting the endpoints of $e_1$ followed by contracting the endpoints of $e_2$. If a cut $(S, S^c)$ does not have $v_0, z_0$ as a singleton on either side, there must be a choice of $e_1, e_2$ that do not cross the cut, which means that the cut "survives" the contraction. Note that the contracted graph is always $(2k - 2)$-edge-connected, so we can proceed as before and produce a list of all of its cuts of size $< 3k$. Taking the union of the list of all such cuts for all choices of $e_1, e_2$ produces the desired list we are seeking. $\square$

We remark that a simple modification of our proof also shows that checking conditions 1 and 2 of Lemma A.8 can be done in polynomial time.

### A.3 Simplification and Details on Lemma 4.5

Here we state the lemma that captures the guarantees of the algorithm $\mathrm{CreateNewProbabilityValues}$ from [ACSS20]. We later apply this lemma in a specific setting where the conditions of Lemma 4.5 are met and provide its proof.

For a given profile $\phi$, the algorithm $\mathrm{CreateNewProbabilityValues}$ takes input $(\mathbf{A}, \mathbf{B}, \mathbf{R})$ and creates a solution pair $(\mathbf{B}', \mathbf{R}')$ that satisfy the following lemma.

**Lemma A.10.** *Given a profile* $\phi \in \Phi^n$ *with* $k$ *distinct frequencies, a probability discretization set* $\mathbf{R}$ *and matrices* $\mathbf{A}, \mathbf{B} \in \mathbb{R}^{[\ell] \times [0,k]}$ *that satisfy:* $\mathbf{A} \in \mathbf{Z}_{\mathbf{R}}^{\phi, frac}$ *and* $\mathbf{B}_{i,j} \leq \mathbf{A}_{i,j}$ *for all* $i \in [\ell]$ *and* $j \in [0, k]$. *There exists an algorithm that outputs a probability discretization set* $\mathbf{R}'$ *and* $\mathbf{A}' \in \mathbb{R}^{[\ell + (k+1)] \times [0,k]}$ *that satisfy the following guarantees,*

1. $\sum_{j\in[0,k]} \boldsymbol{A}'_{i,j} = \sum_{j\in[0,k]} \boldsymbol{B}_{i,j}$ for all $i \in [\ell]$.
2. For any $i \in [\ell+1, \ell+(k+1)]$, let $j \in [0,k]$ be such that $i = \ell+1+j$ then $\boldsymbol{A}'_{\ell+1+j,j'} = 0$ for all $j' \in [0,k]$ and $j' \neq j$. (Diagonal Structure)
3. For any $i \in [\ell+1, \ell+(k+1)]$, let $j \in [0,k]$ be such that $i = \ell+1+j$, then $\sum_{j'\in[0,k]} \boldsymbol{A}'_{i,j'} = \boldsymbol{A}'_{\ell+1+j,j} = \phi_j - \sum_{i'\in[\ell]} \boldsymbol{B}_{i',j}$.
4. $\boldsymbol{A}' \in \boldsymbol{Z}^{\phi,frac}_{\boldsymbol{R}'}$ and $\sum_{i\in[\ell+(k+1)]}\sum_{j\in[0,k]} \boldsymbol{A}'_{i,j} = \sum_{i\in[\ell]}\sum_{j\in[0,k]} \boldsymbol{A}_{i,j}$.
5. Let $\alpha_i \stackrel{\text{def}}{=} \sum_{j\in[0,k]} \boldsymbol{A}_{i,j} - \sum_{j\in[0,k]} \boldsymbol{B}_{i,j}$ for all $i \in [\ell]$ and $\Delta \stackrel{\text{def}}{=} \max(\sum_{i\in[\ell]}(\boldsymbol{A}\overrightarrow{1})_i, \ell \times k)$, then
$$\boldsymbol{g}(\boldsymbol{A}') \geq \exp\left(-O\left(\sum_{i\in[\ell]} \alpha_i \log \Delta\right)\right) \boldsymbol{g}(\boldsymbol{A}).$$
6. For any $j \in [0,k]$, the new level sets have probability value equal to, $\boldsymbol{r}_{\ell+1+j} = \frac{\sum_{i\in[1,\ell]}(\boldsymbol{A}_{ij}-\boldsymbol{B}_{ij})\boldsymbol{r}_i}{\sum_{i\in[1,\ell]}(\boldsymbol{A}_{ij}-\boldsymbol{B}_{ij})}$.

W are now ready to provide the proof of Lemma 4.5

*Proof of Lemma 4.5.* By Lemma A.10, we get a matrix $\boldsymbol{A}' \in \mathbb{R}^{[\ell+(k+1)]\times[0,k]}$ that satisfies $\boldsymbol{A}' \in \boldsymbol{Z}^{\phi,frac}_{\boldsymbol{R}'}$ (guarantee 4 in Lemma A.10) and $\boldsymbol{g}(\boldsymbol{A}') \geq \exp\left(-O\left(\sum_{i\in[\ell]} \alpha_i \log \Delta\right)\right) \boldsymbol{g}(\boldsymbol{A})$, where $\alpha_i \stackrel{\text{def}}{=} \sum_{j\in[0,k]} \boldsymbol{A}_{i,j} - \sum_{j\in[0,k]} \boldsymbol{B}_{i,j}$ for all $i \in [\ell]$ and $\Delta \stackrel{\text{def}}{=} \max(\sum_{i\in[\ell]}(\boldsymbol{A}\overrightarrow{1})_i, \ell \times k)$.

To prove the lemma we need to show two things: $\boldsymbol{A}' \in \boldsymbol{Z}^{\phi}_{\boldsymbol{R}'}$ and $\boldsymbol{g}(\boldsymbol{A}') \geq \exp\left(-O\left(t\log n\right)\right) \boldsymbol{g}(\boldsymbol{A})$. We start with the proof of the first expression. Note that $\boldsymbol{A}' \in \boldsymbol{Z}^{\phi,frac}_{\boldsymbol{R}'}$ and we need to show that $\boldsymbol{A}'$ has all integral row sums. For $i \in [\ell]$, the $i$'th row sum, that is $[\boldsymbol{A}'\mathbf{1}]_i$ is integral by combining guarantee 1 of Lemma A.10 and $[\boldsymbol{B}\mathbf{1}]_i \in \mathbb{Z}_+$ (condition of our current lemma). For $i \in [\ell+1, \ell+(k+1)]$, $[\boldsymbol{A}'\mathbf{1}]_i = \phi_j - [\boldsymbol{B}^\top\mathbf{1}]_j$ (guarantee 3 of Lemma A.10) and the $i$'th row sum is integral because $[\boldsymbol{B}^\top\mathbf{1}]_j \in \mathbb{Z}_+$ (condition of our current lemma) and $[\boldsymbol{B}^\top\mathbf{1}]_j \leq [\boldsymbol{A}^\top\mathbf{1}]_j \leq \phi_j$.

We now shift our attention to the second expression, that is $\boldsymbol{g}(\boldsymbol{A}') \geq \exp\left(-O\left(t\log n\right)\right) \boldsymbol{g}(\boldsymbol{A})$. We prove this inequality by providing bounds on the parameters $\Delta, \alpha_i$. Observe that $\Delta \leq 1/\boldsymbol{r}_{min} + \ell k \leq 1/\boldsymbol{r}_{min} + k(k+1) \leq O(n^2)$ because $\boldsymbol{A} \in \boldsymbol{Z}^{\phi,frac}_{\boldsymbol{R}}$ and therefore satisfies $\sum_{i\in[1,k+1]} \boldsymbol{r}_i[\boldsymbol{A}\mathbf{1}]_i \leq 1$ that further implies $\sum_{i\in[1,k+1]}[\boldsymbol{A}'\mathbf{1}]_i \leq 1/\boldsymbol{r}_{min} \leq 2n^2$ (see the definition of probability discretization). In the second inequality for the bound on $\Delta$ we used $\ell \leq k+1$, as without loss of generality the number of probability values in $|\boldsymbol{R}|$ can be assumed to be at most $k+1$ (because of the sparsity lemma Lemma 4.3) and the actual size of $|\boldsymbol{R}|$ only reflects in the running time. Now note that $\sum_{i\in[k+1]} \alpha_i = \sum_{i\in[\ell],j\in[0,k]}(\boldsymbol{A}_{ij} - \boldsymbol{B}_{ij}) \leq t$ because of the condition of the lemma. Combining the analysis for $\Delta$ and $\alpha_i$, we get $\boldsymbol{g}(\boldsymbol{A}') \geq \exp\left(-O\left(t\log n\right)\right) \boldsymbol{g}(\boldsymbol{A})$ and we conclude the proof. $\qquad\square$

## A.4 Proof of Theorem 4.1 and Theorem 2.1

Here we provide the proof of Theorem 4.1, that provides the guarantees of our first rounding algorithm (Algorithm 1) for any probability descritization set $\boldsymbol{R}$. Later we choose this discretization set carefully to prove our main theorem (Theorem 2.1).

*Proof of Theorem 4.1.* By Lemma 4.2, the Step 1 returns a solution $\boldsymbol{S}' \in \boldsymbol{Z}^{\phi,frac}_{\boldsymbol{R}}$ that satisfies, $C_\phi \cdot \boldsymbol{g}(\boldsymbol{S}') \geq \exp\left(O\left(-k\log n\right)\right) \max_{\boldsymbol{q}\in\Delta^{\mathcal{P}}_{\boldsymbol{R}}} \mathbb{P}(\boldsymbol{q}, \phi)$. By Lemma 4.3, the Step 2 takes input $\boldsymbol{S}'$ and outputs $\boldsymbol{S}'' \in \boldsymbol{Z}^{\phi,frac}_{\boldsymbol{R}}$ such that $\boldsymbol{g}(\boldsymbol{S}'') \geq \boldsymbol{g}(\boldsymbol{S}')$ and $\left|\{i \in [\ell] \mid [\boldsymbol{S}''\overrightarrow{1}]_i > 0\}\right| \leq k+1$. As the matrix $\boldsymbol{S}''$ has at most $k+1$ non-zero rows and columns, by Theorem 4.4 the Step 3 returns a matrix $\boldsymbol{B}''$ that satisfies: $\boldsymbol{B}''_{ij} \leq \boldsymbol{S}''_{ij} \,\forall\, i \in [\ell], j \in [0,k]$, $\boldsymbol{B}''\overrightarrow{1} \in \mathbb{Z}^{\ell}_+$, $\boldsymbol{B}''^\top\overrightarrow{1} \in \mathbb{Z}^{[0,k]}_+$ and $\sum_{i\in[\ell],j\in[0,k]}(\boldsymbol{S}''_{ij} - \boldsymbol{B}''_{ij}) \leq O(k)$. The matrices $\boldsymbol{S}''$ and $\boldsymbol{B}''$ satisfy the conditions of Lemma 4.5 with parameter $t = O(k)$ and the algorithm CreateNewProbabilityValues returns a solution $(\boldsymbol{S}^{\text{ext}}, \boldsymbol{R}^{\text{ext}})$ such that $\boldsymbol{S}^{\text{ext}} \in \boldsymbol{Z}^{\phi}_{\boldsymbol{R}^{\text{ext}}}$ and $\boldsymbol{g}(\boldsymbol{S}^{\text{ext}}) \geq \exp(-O(k\log n))\boldsymbol{g}(\boldsymbol{S}'')$. Further substituting $\boldsymbol{g}(\boldsymbol{S}'') \geq \boldsymbol{g}(\boldsymbol{S}')$ from earlier (Step 2) we get, $\boldsymbol{g}(\boldsymbol{S}^{\text{ext}}) \geq \exp(-O(k\log n))\boldsymbol{g}(\boldsymbol{S}')$. As $\boldsymbol{S}^{\text{ext}} \in \boldsymbol{Z}^{\phi}_{\boldsymbol{R}^{\text{ext}}}$, by Lemma 3.4 the associated distribution $\boldsymbol{p}'$ satisfies $\mathbb{P}(\boldsymbol{p}', \phi) \geq \exp(-O(k\log n))C_\phi \cdot \boldsymbol{g}(\boldsymbol{S}^{\text{ext}}) \geq \exp(-O(k\log n))C_\phi \cdot \boldsymbol{g}(\boldsymbol{S}')$. Further

combined with inequality $C_\phi \cdot \mathbf{g}(\mathbf{S}') \geq \exp\left(O\left(-k \log n\right)\right) \max_{\mathbf{q} \in \Delta_{\mathbf{R}}^{\mathcal{D}}} \mathbb{P}(\mathbf{q}, \phi)$ (Step 1) we get,

$$\mathbb{P}(\mathbf{p}', \phi) \geq \exp\left(O\left(-k \log n\right)\right) \max_{\mathbf{q} \in \Delta_{\mathbf{R}}^{\mathcal{D}}} \mathbb{P}(\mathbf{q}, \phi) .$$

All the steps in our algorithm run in polynomial time and we conclude the proof. □

*Proof of Theorem 2.1.* Choose $\mathbf{R}$ with parameters $\alpha = k \log n / n$ and $|\mathbf{R}| = \ell = O(n/k)$ in Lemma 3.1 and we get that $\max_{\mathbf{q} \in \Delta_{\mathbf{R}}^{\mathcal{D}}} \mathbb{P}(\mathbf{q}, \phi) \geq \exp\left(-k \log n\right) \max_{\mathbf{p} \in \Delta^{\mathcal{D}}} \mathbb{P}(\mathbf{p}, \phi)$. As the $|\mathbf{R}|$ is polynomial in $n$, the previous inequality combined with Theorem 4.1 proves our theorem. □

# B PseudoPML Approach, Remaining Proofs from Section 5 and Experiments

Here we provide all the details regarding the PseudoPML approach. PseudoPML also known as TrucatedPML was introduced independently in [CSS19b] and [HO19]. In Appendix B.1, we provide the proof for the guarantees achieved by our second rounding algorithm (Theorem 5.1) that in turn helps us prove Theorem 2.4. In Appendix B.2, we provide notations and definitions related to the PseudoPML approach. In Appendix B.3, we provide the proof of Lemma 2.3. Finally in Appendix B.4, we provide the remaining experimental results and the details of our implementation.

## B.1 Proof of Theorem 5.1 and Theorem 2.4

Here we provide the proof of Theorem 5.1 that provides the guarantees satisfied by our second approximate PML algorithm. Further using this theorem , we provide the proof for Theorem 2.4.

*Proof of Theorem 5.1.* By Lemma 4.2, the first part of Step 1 returns a solution $\mathbf{X} \in \mathbf{Z}_{\mathbf{R}}^{\phi, frac}$ that satisfies,
$$C_\phi \cdot \mathbf{g}(\mathbf{X}) \geq \exp\left(O\left(-k \log n\right)\right) \max_{\mathbf{q} \in \Delta_{\mathbf{R}}^{\mathcal{D}}} \mathbb{P}(\mathbf{q}, \phi) . \tag{10}$$

We also sparsify the solution $\mathbf{X}$ in Step 1 that we call $\mathbf{X}'$. By Lemma 4.3, the solution $\mathbf{X}' \in \mathbf{Z}_{\mathbf{R}}^{\phi, frac}$ satisfies $\mathbf{g}(\mathbf{X}') \geq \mathbf{g}(\mathbf{X})$ and $\left|\{i \in [\ell] \mid [\mathbf{X}'\overrightarrow{1}]_i > 0\}\right| \leq k + 1$. The Steps 2-3 of our algorithm throw away the zero rows of matrix $\mathbf{X}'$ and consider the sub matrix $\mathbf{S}'$ corresponding to its non-zeros rows. Let $\mathbf{R}'$ be the probability values that correspond to these non-zero rows of $\mathbf{X}'$ and $\mathbf{S}' \in \mathbf{Z}_{\mathbf{R}'}^{\phi, frac}$. As $\mathbf{S}'$ changes during Steps 4-8 of the algorithm, we use $\mathbf{Y}$ to denote the unchanged $\mathbf{S}'$ from Step 2. The matrix $\mathbf{Y} \in \mathbf{Z}_{\mathbf{R}'}^{\phi, frac}$ satisfies: $\mathbf{g}(\mathbf{Y}) = \mathbf{g}(\mathbf{X}') \geq \mathbf{g}(\mathbf{X})$ and has $\ell' \leq k + 1$ rows. In the remainder of the proof we show that the distribution $\mathbf{p}'$ outputted by our algorithm satisfies $\mathbb{P}(\mathbf{p}', \phi) \geq \exp\left(-O((\mathbf{r}_{\max} - \mathbf{r}_{min})n + k \log(\ell n))\right) C_\phi \cdot \mathbf{g}(\mathbf{Y})$ that further combined with $\mathbf{g}(\mathbf{Y}) \geq \mathbf{g}(\mathbf{X})$ and Equation (10) proves the theorem. Now recall the definition of $\mathbf{g}(\mathbf{Y})$,

$$\mathbf{g}(\mathbf{Y}) \stackrel{\text{def}}{=} \exp\left( \sum_{i \in [1, \ell'], j \in [0, k]} \left[\mathbf{C}'_{ij} \mathbf{Y}_{ij} - \mathbf{Y}_{ij} \log \mathbf{Y}_{ij}\right] + \sum_{i \in [1, \ell']} [\mathbf{Y1}]_i \log[\mathbf{Y1}]_i \right), \tag{11}$$

where $\mathbf{C}'_{ij} = \mathbf{m}_j \log \mathbf{r}'_i$. We refer to the linear term in $\mathbf{Y}$ of function $\mathbf{g}(\mathbf{Y})$ as the first term and the remaining entropy like terms as the second. We denote the elements of set $\mathbf{R}'$ by $\mathbf{r}'_i$ and let $\mathbf{r}'_1 < \ldots \mathbf{r}'_{\ell'}$. The Steps 4-8 of our rounding algorithm transfer the mass of $\mathbf{S}'$ from lower probability value rows to higher ones while maintaining the integral row sum for the current row . Formally at iteration $i$, our algorithm takes the current fractional part of the $i$'th row sum $([\mathbf{S}'\mathbf{1}]_i - \lfloor[\mathbf{S}'\mathbf{1}]_i\rfloor)$ and moves it to row $i + 1$ (corresponding to higher probability value) by updating matrix $\mathbf{S}'$. As the first term in function $\mathbf{g}(\cdot)$ is strictly increasing in the values of $\mathbf{r}'_i$, it is immediate that the final solution $\mathbf{S}^{\text{ext}}$ satisfies,

$$\sum_{i \in [1, \ell'], j \in [0, k]} \mathbf{C}'_{ij} \mathbf{S}^{\text{ext}}_{ij} \geq \sum_{i \in [1, \ell'], j \in [0, k]} \mathbf{C}'_{ij} \mathbf{Y}_{ij} . \tag{12}$$

The movement of the mass between the rows happen within the same column, therefore $\mathbf{S}^{\text{ext}}$ satisfies the column constraints, that is $[\mathbf{S}^{\text{ext}\top}\mathbf{1}]_j = \phi_j$ for all $j \in [k]$. As $[\mathbf{S}^{\text{ext}}\mathbf{1}]_i = \lfloor[\mathbf{S}'\mathbf{1}]_i\rfloor$ for all $i \in [1, \ell]$,

we also have that all the row sums are integral. Therefore to prove the theorem all that remains is to bound the loss in objective corresponding to the second term for Steps 4-8 and analysis of Steps 9-11.

In Steps 4-8 at iteration $i$, note that we move at most 1 unit of mass $(\frac{\lfloor[\mathbf{S}'\mathbf{1}]_i\rfloor}{[\mathbf{S}'\mathbf{1}]_i})$ from row $i$ to $i+1$. Therefore the updated matrix $\mathbf{S}'$ after Step 6 satisfies $\sum_{j\in[0,k]}(\mathbf{S}'_{i+1,j} - \mathbf{Y}_{i+1,j}) \leq 1$. As $\mathbf{S}^{\text{ext}}_{i+1,j} = \mathbf{S}'_{i+1,j}\frac{\lfloor\|\mathbf{S}'_{i+1}\|_1\rfloor}{\|\mathbf{S}'_{i+1}\|_1}$ we have $\sum_{j\in[0,k]}(\mathbf{S}'_{i+1,j} - \mathbf{S}^{\text{ext}}_{i+1,j}) \leq 1$ and further combined with the previous inequality we get $\sum_{j\in[0,k]}|\mathbf{S}^{\text{ext}}_{i+1,j} - \mathbf{Y}_{i+1,j}| \leq 1$ for all $i \in [1, \ell'-1]$. For the first row, we have $\mathbf{S}^{\text{ext}}_{1,j} = \mathbf{Y}_{1,j}\frac{\lfloor\|\mathbf{Y}_1\|_1\rfloor}{\|\mathbf{Y}_1\|_1}$ which also gives $\sum_{j\in[0,k]}|\mathbf{S}^{\text{ext}}_{1,j} - \mathbf{Y}_{1,j}| \leq 1$. Therefore for all $i \in [1, \ell']$ the following inequality holds,

$$\sum_{j\in[0,k]} |\mathbf{S}^{\text{ext}}_{i,j} - \mathbf{Y}_{i,j}| \leq 1 \,. \tag{13}$$

As the function $x\log x$ and $-x\log x$ are $O(\log n)$-Lipschitz when $x \in [\frac{1}{n^{10}}, \infty] \cup \{0\}$ and all the terms where $\mathbf{Y}_{i,j}, [\mathbf{Y}\mathbf{1}]_i, \mathbf{S}^{\text{ext}}_{i,j}, [\mathbf{S}^{\text{ext}}\mathbf{1}]_i$ take values less than $1/n^{10}$ contribute very little (at most $\exp(O(1/n^8))$) to the objective. Therefore by Equation (13) we get,

$$\sum_{i\in[1,\ell'],j\in[0,k]} \left(-\mathbf{S}^{\text{ext}}_{ij}\log\mathbf{S}^{\text{ext}}_{ij}\right) \geq \sum_{i\in[1,\ell'],j\in[0,k]} \left(-\mathbf{Y}_{ij}\log\mathbf{Y}_{ij}\right) - O(\ell'\log n) \,, \tag{14}$$

$$\sum_{i\in[1,\ell']} [\mathbf{S}^{\text{ext}}\mathbf{1}]_i\log[\mathbf{S}^{\text{ext}}\mathbf{1}]_i \geq \sum_{i\in[1,\ell']} [\mathbf{Y}\mathbf{1}]_i\log[\mathbf{Y}\mathbf{1}]_i - O(\ell'\log n) \,, \tag{15}$$

where in the above inequalities we used the Lipschitzness of entropy and negative of entropy functions. Therefore Steps 4-8 of the algorithm outputs a solution $\mathbf{S}^{\text{ext}}$ that along with other conditions also satisfies Equations (12), (14) and (15). Now observe that we are not done yet as the solution $\mathbf{S}^{\text{ext}}$ might violate the distributional constraint $\sum_{i\in[1,\ell']} \mathbf{r}'_i\|\mathbf{S}^{\text{ext}}_i\|_1 \leq 1$; to address this in Steps 9-10 we construct a new probability $\mathbf{R}^{\text{ext}}$ where we scale down the probability values in $\mathbf{R}'$ by $c = \sum_{i\in[1,\ell']} \mathbf{r}'_i\|\mathbf{S}^{\text{ext}}_i\|_1$. Such a scaling immediately ensures the satisfaction of the distributional constraint with respect to $\mathbf{R}^{\text{ext}}$. As the row sums of $\mathbf{S}^{\text{ext}}$ are integral and it satisfies all the column constraints as well, we have that $\mathbf{S}^{\text{ext}} \in \mathbf{Z}^{\phi}_{\mathbf{R}^{\text{ext}}}$. Let $\mathbf{r}''_i = \mathbf{r}'_i/c$ be the probability values in set $\mathbf{R}^{\text{ext}}$, then note that,

$$\begin{aligned}
\sum_{i\in[1,\ell'],j\in[0,k]} \mathbf{m}_j\mathbf{S}^{\text{ext}}_{ij}\log\mathbf{r}''_i &= \sum_{i\in[1,\ell'],j\in[0,k]} \mathbf{m}_j\mathbf{S}^{\text{ext}}_{ij}\log\frac{\mathbf{r}'_i}{c} \\
&= \sum_{i\in[1,\ell'],j\in[0,k]} \mathbf{C}'_{i,j}\mathbf{S}^{\text{ext}}_{ij} - \log c \sum_{i\in[1,\ell'],j\in[0,k]} \mathbf{m}_j\mathbf{S}^{\text{ext}}_{ij} \\
&= \sum_{i\in[1,\ell'],j\in[0,k]} \mathbf{C}'_{i,j}\mathbf{S}^{\text{ext}}_{ij} - \log c \sum_{j\in[0,k]} \mathbf{m}_j\phi_j \\
&= \sum_{i\in[1,\ell'],j\in[0,k]} \mathbf{C}'_{i,j}\mathbf{S}^{\text{ext}}_{ij} - n\log c \,.
\end{aligned} \tag{16}$$

All that remains is to provide an upper bound on the value of $c$. Observe that, $c = \sum_{i\in[1,\ell']} \mathbf{r}'_i\|\mathbf{S}^{\text{ext}}_i\|_1 = \sum_{i\in[1,\ell']} \mathbf{r}'_i\|\mathbf{Y}_i\|_1 + \sum_{i\in[1,\ell']} \mathbf{r}'_i(\|\mathbf{S}^{\text{ext}}_i\|_1 - \|\mathbf{Y}_i\|_1) \leq 1 + \mathbf{r}_{\max} - \mathbf{r}_{min}$, where in the last inequality we used $\mathbf{Y} \in \mathbf{Z}^{\phi}_{\mathbf{R}'}$ and $\sum_{i\in[1,\ell']}(\|\mathbf{S}^{\text{ext}}_i\|_1 - \|\mathbf{Y}_i\|_1) = 0$. Substituting the bound on $c$ back into Equation (16) we get,

$$\begin{aligned}
\sum_{i\in[1,\ell'],j\in[0,k]} \mathbf{m}_j\mathbf{S}^{\text{ext}}_{ij}\log\mathbf{r}''_i &= \sum_{i\in[1,\ell'],j\in[0,k]} \mathbf{C}'_{i,j}\mathbf{S}^{\text{ext}}_{ij} - n\log c \\
&\geq \sum_{i\in[1,\ell'],j\in[0,k]} \mathbf{C}'_{i,j}\mathbf{S}^{\text{ext}}_{ij} - O((\mathbf{r}_{\max} - \mathbf{r}_{min})n) \,.
\end{aligned} \tag{17}$$

Using Equations (12), (14), (15) and (17), the function value $\mathbf{g}(\mathbf{S}^{\text{ext}})$ with respect to $\mathbf{R}^{\text{ext}}$ satisfies,

$$\begin{aligned}
\mathbf{g}(\mathbf{S}^{\text{ext}}) &\geq \exp\left(-O(\mathbf{r}_{\max} - \mathbf{r}_{min})n - O(\ell'\log n)\right)\mathbf{g}(\mathbf{Y}) \\
&\geq \exp\left(-O(\mathbf{r}_{\max} - \mathbf{r}_{min})n - O(k\log n)\right)\mathbf{g}(\mathbf{Y}),
\end{aligned} \tag{18}$$

where in the last inequality we used $\ell' \leq k+1$. As $\mathbf{S}^{\text{ext}} \in \mathbf{Z}_{\mathbf{R}^{\text{ext}}}^{\phi}$, by Lemma 3.4 the associated distribution $\mathbf{p}'$ satisfies $\mathbb{P}(\mathbf{p}', \phi) \geq \exp(-O(k \log n))C_\phi \cdot \mathbf{g}(\mathbf{S}^{\text{ext}})$. Further combined with Equation (18), $\mathbf{g}(\mathbf{Y}) \geq \mathbf{g}(\mathbf{X})$ and Equation (10) we get,

$$\mathbb{P}(\mathbf{p}', \phi) \geq \exp\left(-O(\mathbf{r}_{\max} - \mathbf{r}_{min})n - O(k \log n)\right) \max_{\mathbf{q} \in \Delta_\mathbf{R}^\mathcal{D}} \mathbb{P}(\mathbf{q}, \phi) .$$

In the remainder we provide the analysis for the running time of our algorithm. By Theorem A.2 we can solve the convex optimization problem in Step 1 in time $\widetilde{O}(|\mathbf{R}|k^2)$. By Lemma 4.3, the sub routine Sparse can be implemented in time $\widetilde{O}(|\mathbf{R}|k^\omega)$ and all the remaining steps correspond to the low order terms; therefore the final run time of our algorithm is $\widetilde{O}(|\mathbf{R}|k^\omega)$ and we conclude the proof. $\qquad\square$

The above result holds for a general $\mathbf{R}$ and we choose this set carefully to prove Theorem 2.4.

*Proof of Theorem 2.4.* As the probability values lie in a restricted range, we just need to discretize the interval $[\ell, u]$. We choose the probability discretization set $\mathbf{R}$ with parameters $\alpha = k/n$, $\mathbf{r}_{\max} = u$, $\mathbf{r}_{min} = \ell$ and $|\mathbf{R}| = O(\frac{n \log \frac{u}{\ell}}{k})$. By Lemma 3.1, we have $\max_{\mathbf{q} \in \Delta_\mathbf{R}^\mathcal{D}} \mathbb{P}(\mathbf{q}, \phi) \geq \exp(-k-6)\mathbb{P}(\mathbf{p}, \phi)$. Further combined with Theorem 5.1, we conclude our proof. $\qquad\square$

## B.2 Notation and the General Framework

Here we provide all the definitions and description of the general framework for symmetric property estimation using the PseudoPML [CSS19b, HO19]. We start by providing definitions of pseudo profile and PseudoPML distributions.

**Definition B.1** (*S*-pseudo Profile). For any sequence $y^n \in \mathcal{D}^n$ and $S \subseteq \mathcal{D}$, let $\mathbf{M} \stackrel{\text{def}}{=} \{\mathbf{f}(y^n, x)\}_{x \in S}$ be the set of distinct frequencies from $S$ and let $\mathbf{m}_1, \mathbf{m}_2, \ldots, \mathbf{m}_{|\mathbf{M}|}$ be these distinct frequencies. The *S-pseudo* profile of a sequence $y^n$ and set $S$ denoted by $\phi_S = \Phi_S(y^n)$ is a vector in $\mathbb{Z}_+^{|\mathbf{M}|}$, where $\phi_S(j) \stackrel{\text{def}}{=} |\{x \in S \mid \mathbf{f}(y^n, x) = \mathbf{m}_j\}|$ is the number of domain elements in $S$ with frequency $\mathbf{m}_j$. We call $n$ the length of $\phi_S$ as it represents the length of the sequence $y^n$ from which the pseudo profile was constructed. Let $\Phi_S^n$ denote the set of all *S*-pseudo profiles of length $n$.

The probability of a *S*-pseudo profile $\phi_S \in \Phi_S^n$ with respect to $\mathbf{p} \in \Delta^\mathcal{D}$ is defined as follows,

$$\Pr(\mathbf{p}, \phi_S) \stackrel{\text{def}}{=} \sum_{\{y^n \in \mathcal{D}^n \mid \Phi_S(y^n) = \phi_S\}} \mathbb{P}(\mathbf{p}, y^n), \tag{19}$$

we use notation $\Pr$ instead of $\mathbb{P}$ to differentiate between the probability of a pseudo profile from the profile.

**Definition B.2** (*S*-PseudoPML distribution). For any *S*-pseudo profile $\phi_S \in \Phi_S^n$, a distribution $\mathbf{p}_{\phi_S} \in \Delta^\mathcal{D}$ is a *S-PseudoPML* distribution if $\mathbf{p}_{\phi_S} \in \arg\max_{\mathbf{p} \in \Delta^\mathcal{D}} \mathbb{P}(\mathbf{p}, \phi_S)$. Further, a distribution $\mathbf{p}_{\phi_S}^\beta \in \Delta^\mathcal{D}$ is a $(\beta, S)$-*approximate PseudoPML* distribution if $\mathbb{P}(\mathbf{p}_{\phi_S}^\beta, \phi_S) \geq \beta \cdot \mathbb{P}(\mathbf{p}_{\phi_S}, \phi_S)$.

We next provide the description of the general framework from [CSS19b]. The input to this general framework is a sequence of $2n$ i.i.d sample denoted by $x^{2n}$ from an underlying hidden distribution $\mathbf{p}$, a symmetric property of interest $\mathbf{f}$ and a set of frequencies F. The output is an estimate of $f(\mathbf{p})$ using a mixture of PML and empirical distributions.

---

**Algorithm 3** General Framework for Symmetric Property Estimation

---

1: **procedure** PROPERTY ESTIMATION($x^{2n}, \mathbf{f}, \text{F}$)
2:     Let $x^{2n} = (x_1^n, x_2^n)$, where $x_1^n$ and $x_2^n$ represent first and last $n$ samples of $x^{2n}$ respectively.
3:     Define $S \stackrel{\text{def}}{=} \{y \in \mathcal{D} \mid f(x_1^n, y) \in \text{F}\}$.
4:     Construct profile $\phi_S$, where $\phi_S(j) \stackrel{\text{def}}{=} |\{y \in S \mid \mathbf{f}(x_2^n, y) = j\}|$.
5:     Find a $(\beta, S)$-approximate PseudoPML distribution $\mathbf{p}_{\phi_S}^\beta$ and empirical distribution $\hat{\mathbf{p}}$ on $x_2^n$.
6:     **return** $\mathbf{f}_S(\mathbf{p}_{\phi_S}^\beta) + \mathbf{f}_{\bar{S}}(\hat{\mathbf{p}})$ + correction bias with respect to $\mathbf{f}_{\bar{S}}(\hat{\mathbf{p}})$.
7: **end procedure**

---

We call the procedure of estimation using the above general framework as the PseudoPML approach.

## B.3  Proof of Lemma 2.3 and the Implementation of General Framework

Here we provide the proof of Lemma 2.3. The main idea behind the proof of this lemma is to use an efficient solver for the computation of approximate PML to return an approximate PseudoPML distribution. The following lemma will be useful to establish such a connection and we define the following notations: $\Delta^S_{[\ell,u]} \overset{\text{def}}{=} \{\mathbf{p} \in \Delta^S \big| \mathbf{p}_x \in [\ell, u] \ \forall x \in S\}$ and further define $\Delta^{\mathcal{D}}_{S,[\ell,u]} \overset{\text{def}}{=} \{\mathbf{p} \in \Delta^{\mathcal{D}} \big| \mathbf{p}_x \in [\ell, u] \ \forall x \in S\}$, where $\Delta^S$ are all distributions that are supported on domain $S$.

**Lemma B.3.** *For any profile $\phi' \in \Phi^{n'}$ with $k'$ distinct frequencies, domain $S \subset \mathcal{D}$ and $\ell', u' \in [0, 1]$. If there is an algorithm that runs in time $T(n', k', u', \ell')$ and returns a distribution $\mathbf{p}' \in \Delta^S$ such that,*

$$\mathbb{P}(\mathbf{p}', \phi') \geq \exp\left(-O((u' - \ell')n' \log n' + k' \log n')\right) \max_{\mathbf{q} \in \Delta^S_{[\ell',u']}} \mathbb{P}(\mathbf{q}, \phi') .$$

*Then for domain $\mathcal{D}$, any pseudo $\phi_S \in \Phi^n_S$ with $k$ distinct frequencies and $\ell, u \in [0, 1]$, such an algorithm can be used to compute $\mathbf{p}''_S$, part corresponding to $S \subseteq \mathcal{D}$ of distribution $\mathbf{p}'' \in \Delta^{\mathcal{D}}$ in time $T(n, k, u, \ell)$ where the distribution $\mathbf{p}''$ further satisfies,*

$$\Pr(\mathbf{p}'', \phi_S) \geq \exp\left(-O((u - \ell)n \log n + k \log n)\right) \max_{\mathbf{q} \in \Delta^{\mathcal{D}}_{S,[\ell,u]}} \Pr(\mathbf{q}, \phi_S) .$$

*Proof.* Recall that,

$$\Pr(\mathbf{q}, \phi_S) \overset{\text{def}}{=} \sum_{\{y^n \in \mathcal{D}^n \mid \Phi_S(y^n) = \phi_S\}} \mathbb{P}(\mathbf{q}, y^n) .$$

Let $\mathbf{q}_S$ and $\mathbf{q}_{\bar{S}}$ denote the part of distribution $\mathbf{q}$ corresponding to $S, \bar{S} \subseteq \mathcal{D}$; they are pseudo distributions supported on $S$ and $\bar{S}$ respectively. Let $n_1 = \sum_{\mathbf{m}_j \in \phi_S} \mathbf{m}_j$ and $n_2 \overset{\text{def}}{=} \sum_{\mathbf{m}_j \in \phi_{\bar{S}}} \mathbf{m}_j$ then,

$$\mathbb{P}(\mathbf{q}_S, \phi_S) \overset{\text{def}}{=} \sum_{\{y^{n_1} \in S^{n_1} \mid \Phi(y^{n_1}) = \phi_S\}} \prod_{x \in S} \mathbf{q}_x^{\mathbf{f}(y^{n_1}, x)}$$

$$\mathbb{P}(\mathbf{q}_{\bar{S}}, \phi_{\bar{S}}) \overset{\text{def}}{=} \sum_{\{y^{n_2} \in \bar{S}^{n_2} \mid \Phi(y^{n_2}) = \phi_{\bar{S}}\}} \prod_{x \in \bar{S}} \mathbf{q}_x^{\mathbf{f}(y^{n_2}, x)}$$

We can write the probability of a pseudo profile in terms of the above functions as follows,

$$\Pr(\mathbf{q}, \phi_S) = \mathbb{P}(\mathbf{q}_S, \phi_S)\mathbb{P}(\mathbf{q}_{\bar{S}}, \phi_{\bar{S}}).$$

Therefore,

$$\max_{\mathbf{q} \in \Delta^{\mathcal{D}}} \Pr(\mathbf{q}, \phi_S) = \max_{\mathbf{q} \in \Delta^{\mathcal{D}}} \mathbb{P}(\mathbf{q}_S, \phi_S)\mathbb{P}(\mathbf{q}_{\bar{S}}, \phi_{\bar{S}}) ,$$

In the applications of PseudoPML, we just require the part of the distribution corresponding to $S \subseteq \mathcal{D}$ and in the remainder we focus on its computation by exploiting the product structure in the objective.

$$\max_{\mathbf{q} \in \Delta^{\mathcal{D}}} \mathbb{P}(\mathbf{q}_S, \phi_S)\mathbb{P}(\mathbf{q}_{\bar{S}}, \phi_{\bar{S}}) = \max_{\alpha \in [0,1]} \left(\alpha^{n_1} \max_{\mathbf{q}' \in \Delta^S} \mathbb{P}(\mathbf{q}', \phi_S)\right) \left((1 - \alpha)^{n_2} \max_{\mathbf{q}'' \in \Delta^{\bar{S}}} \mathbb{P}(\mathbf{q}'', \phi_{\bar{S}})\right) ,$$

where in the above objective we converted the terms involving the pseudo distributions to distributions. The above equality holds because scaling all the probability values of a distribution by a factor of $\alpha$ scales the PML objective by a factor of $\alpha$ to the power of length of the profile, which is $n_1$ and $n_2$ for $\phi_S$ and $\phi_{\bar{S}}$ respectively. The above objective is nice as we can just focus on the first term in the objective corresponding to $S$ given the optimal $\alpha$ value. Note in the above optimization problem the terms $\max_{\mathbf{q}' \in \Delta^S} \mathbb{P}(\mathbf{q}', \phi_S)$ and $\max_{\mathbf{q}'' \in \Delta^{\bar{S}}} \mathbb{P}(\mathbf{q}'', \phi_{\bar{S}})$ are independent of $\alpha$ and we can solve for the optimum $\alpha$ by finding the maximizer of the following optimization problem.

$$\max_{\alpha \in [0,1]} \alpha^{n_1} (1 - \alpha)^{n_2} .$$

The above optimization problem has a standard closed form solution and the optimum solution is $\alpha^* = \frac{n_1}{n_1 + n_2} = \frac{n_1}{n}$. To summarize, the part of distribution $\mathbf{p}''$ corresponding to $S$ that satisfies the

guarantees of the lemma can be computed by solving the optimization problem $\max_{\mathbf{q}' \in \Delta^S} \Pr(\mathbf{q}', \phi_S)$ upto multiplicative accuracy of $\exp\left(-O((u - \ell)n \log n + k \log n)\right)$ and then scaling all the entries of the corresponding distribution supported on $S$ by a factor of $n_1/n$; which by the conditions of the lemma can be computed in time $T(n, k, \ell, u)$ and we conclude the proof. $\qquad\square$

Using the above lemma we now provide the proof for Lemma 2.3.

*Proof of Lemma 2.3.* Let $\mathbf{p}, \mathbf{p}^{\beta}_{\phi_S}$ be the underlying hidden distribution and $(\beta, S)$-approximate PseudoPML distribution. The guarantees stated in the lemma are the efficient version of Theorem 3.9 and 3.10 in [CSS19b]. Both these theorems are derived using Theorem 3.8 in [CSS19b] that in turn depends on Theorem 3.7 which captures the performance of an approximate PseudoPML distribution. In all these proofs the only expression where the definition of $(\beta, S)$-approximate PseudoPML distribution was used is the following: $\Pr\left(\mathbf{p}^{\beta}_{\phi_S}, \phi_S\right) \geq \beta \Pr(\mathbf{p}, \phi_S)$. Any other distribution $\mathbf{p}'$ that satisfies $\Pr(\mathbf{p}', \phi_S) \geq \beta \Pr(\mathbf{p}, \phi_S)$ also has the same guarantees and provides the efficient version of Theorem 3.9 and 3.10, that is the guarantees of our lemma.

As described in Appendix B.2, the general framework works in two steps. In the first step, it takes the first half of the samples $(x^n_1)$ and determines the set $S \stackrel{\text{def}}{=} \{y \in \mathcal{D} \mid f(x^n_1, y) \in \mathrm{F}\}$, where $\mathrm{F}$ is a predetermined subset of frequencies (input to the general framework) that depends on the property of interest. The pseudo profile $\phi_S$ is computed on the second half of the samples, that is $\phi_S(j) \stackrel{\text{def}}{=} |\{y \in S \mid \mathbf{f}(x^n_2, y) = j\}|$. Based on the frequency of the elements of $S$ in the first half of the sample (they all belong to $\mathrm{F}$), with high probability (in the number of samples) we have an interval $I = [\ell, u]$ in which all the probability values of elements in $S \subseteq \mathcal{D}$ for $\mathbf{p}$ lie. Therefore finding a distribution $\mathbf{p}'$ that satisfies,

$$\Pr(\mathbf{p}', \phi_S) \geq \beta \max_{\mathbf{q} \in \Delta^{\mathcal{D}}_{S,I}} \Pr(\mathbf{q}, \phi_S) \implies \Pr(\mathbf{p}', \phi_S) \geq \beta \Pr(\mathbf{p}, \phi_S) \ ,$$

where $\Delta^{\mathcal{D}}_{S,I} \stackrel{\text{def}}{=} \left\{\mathbf{q} \in \Delta^{\mathcal{D}} \,\middle|\, \mathbf{q}_x \in I \text{ for all } x \in S\right\}$; therefore $\mathbf{p}'$ can be used as a proxy for $\mathbf{p}^{\beta}_{\phi_S}$ and both these distributions satisfy the guarantees of our lemma (for entropy and distance to uniformity) for an appropriately chosen $\beta$. The value of $\beta$ depends on the size of $\mathrm{F}$ that further depends on the property of interest and we analyze this parameter for each property in the final parts of the proof.

Now note that we need to find a distribution $\mathbf{p}'$ that satisfies, $\Pr(\mathbf{p}', \phi_S) \geq \beta \max_{\mathbf{q} \in \Delta^{\mathcal{D}}_{S,I}} \Pr(\mathbf{p}, \phi_S)$ and to implement the PseudoPML approach all we need is $\mathbf{p}'_S$, the part of the distribution corresponding to $S$. The Lemma B.3 helps reduce the problem of computing PseudoPML to PML and we use the algorithm given to us by the condition of our lemma to compute $\mathbf{p}'_S$.

In the remainder, we study the running time and the value of $\beta$ for entropy and distance to uniformity.

**Entropy:** In the application of general framework (Algorithm 3) to entropy, the authors in [CSS19b] choose $F = [0, c \log n]$, where $c > 0$ is a fixed constant (See proof of Theorem 3.9 in [CSS19b]). Recall the definition of subset $S \stackrel{\text{def}}{=} \{y \in \mathcal{D} \mid f(x^n_1, y) \in \mathrm{F}\}$ and as argued in the proof of Theorem 3.9 in [CSS19b], with high probability all the domain elements $x \in S$ have probability values $\mathbf{p}_x \leq \frac{2c \log n}{n}$. Further, we can assume that the minimum non-zero probability of distribution $\mathbf{p}$ to be $\Omega(1/\text{poly}(n))$, because in our setting $n \in \Omega(N/\log N)$ for all error parameters $\epsilon$ and the probability values less than $1/\text{poly}(n)$ contribute very little to the probability mass or entropy of the distribution and we can ignore them. Therefore to implement the PseudoPML approach for entropy all we need is the part corresponding to $S$ of distribution $\mathbf{p}'$ that satisfies,

$$\Pr(\mathbf{p}', \phi_S) \geq \beta \max_{\mathbf{q} \in \Delta^{\mathcal{D}}_{S,I}} \Pr(\mathbf{q}, \phi_S) \ , \tag{20}$$

for any $\beta > \exp\left(-O(\log^2 n)\right)$ (Theorem 3.9 in [CSS19b]) and $I = [\frac{1}{\text{poly}(n)}, \frac{2c \log n}{n}]$. Based on our discussion at the start of the proof, this corresponds to computing the $\beta$-approximate PML distribution supported on $S$ for the profile $\phi_S$. As the number of distinct frequencies in the profile $\phi_S$ is at most $O(\log n)$, length of the profile $\phi_S$ is at most $n$ and interval $I = [\ell, u]$ take values $\ell = 1/\text{poly}(n)$ and $u = O(\frac{\log n}{n})$, the algorithm given by the conditions of our lemma computes the part corresponding

to $S$ of distribution $\mathbf{p}'$ that satisfies Equation (20) with approximation factor $\beta > \exp\left(-O(\log^2 n)\right)$ in time $T(n, O(\log n), 1/\mathrm{poly}(n), O(\frac{\log n}{n}))$.

The proof for distance to uniformity is similar to that of entropy and is described below.

**Distance to Uniformity:** For distance to uniformity, the authors in [CSS19b] choose $F = [\frac{n}{N} - \sqrt{\frac{cn \log n}{N}}, \frac{n}{N} + \sqrt{\frac{cn \log n}{N}}]$, where $c$ is a fixed constant (See proof of Theorem 3.10 in [CSS19b]). The subset $S \stackrel{\text{def}}{=} \{y \in \mathcal{D} \mid f(x_1^n, y) \in F\}$ and as argued in the proof of Theorem 3.10 in [CSS19b], with high probability all the domain elements $x \in S$ have probability values $\mathbf{p}_x \in [\frac{1}{N} - \sqrt{\frac{2c \log n}{nN}}, \frac{1}{N} + \sqrt{\frac{2c \log n}{nN}}]$. Therefore to implement the PseudoPML approach for distance to uniformity all we need is the part corresponding to $S$ of distribution $\mathbf{p}'$ that satisfies,

$$\Pr(\mathbf{p}', \phi_S) \geq \beta \max_{\mathbf{q} \in \Delta_{S,I}^{\mathcal{D}}} \Pr(\mathbf{q}, \phi_S) , \tag{21}$$

for any $\beta > \exp\left(-O(\sqrt{\frac{cn \log^3 n}{N}})\right)$ (Theorem 3.10 in [CSS19b]) and $I = [\frac{1}{N} - \sqrt{\frac{2c \log n}{nN}}, \frac{1}{N} + \sqrt{\frac{2c \log n}{nN}}]$. This corresponds to computing the $\beta$-approximate PML distribution supported on $S$ for the profile $\phi_S$. As the number of distinct frequencies in the profile $\phi_S$ is at most $\sqrt{\frac{2cn \log n}{N}} \in O(1/\epsilon)$ (because $n = \Theta(\frac{N}{\epsilon^2 \log N})$ for distance to uniformity), length of the profile $\phi_S$ is at most $n$ and interval $I = [\ell, u]$ take values $\ell = \frac{1}{N} - \sqrt{\frac{2c \log n}{nN}} \in \Omega(1/N)$ and $u = \frac{1}{N} + \sqrt{\frac{2c \log n}{nN}} \in O(1/N)$, the algorithm given by the conditions of our lemma computes the part corresponding to $S$ of distribution $\mathbf{p}'$ that satisfies Equation (21) with approximation factor $\beta > \exp\left(-O(\sqrt{\frac{cn \log^3 n}{N}})\right)$ in time $T(n, O(1/\epsilon), \Omega(1/N), O(1/N))$. We conclude the proof. $\qquad\square$

## B.4 Experiments

In this section, we provide details related to PseudoPML implementation and some additional experiments. We perform different sets of experiments for entropy estimation – first to compare performance guarantees of PseudoPML approach implemented using our rounding algorithm to the other state-of-the-art estimators and the other to compare the performance of the PseudoPML approach implemented using our approximate PML algorithm (Algorithm 2) with a heuristic algorithm [PJW17].

All the plots in this section depict the performance of various algorithms for estimating entropy of different distributions with domain size $N = 10^5$. Each data point represents 50 random trials. "Uniform" is the uniform distribution, "Mix 2 Uniforms" is a mixture of two uniform distributions, with half the probability mass on the first $N/10$ symbols and the remaining mass on the last $9N/10$ symbols, and $\mathrm{Zipf}(\alpha) \sim 1/i^\alpha$ with $i \in [N]$. In the PseudoPML implementation for entropy, we divide the samples into two parts. We run the empirical estimate on one (this is easy) and the PML estimate on the other. Similar to [CSS19b], we pick $threshold = 18$ (same as [WY16a]) to divide the samples, i.e. we use the PML estimate on frequencies $\leq 18$ and empirical estimate on the rest. As in [CSS19b], we do not perform sample splitting. In all the plots, "Our work" corresponds to the implementation of this PseudoPML approach using our second approximate PML algorithm presented in Section 5 (Algorithm 2). Refer to [CSS19b] for further details on the PseudoPML approach.

In Figure 2, we compare performance guarantees of our work to the other state-of-the-art estimators for entropy. We already did this comparison in Section 5.1 and here we do it for three other distributions. As described in Section 5.1, MLE is the naive approach of using the empirical distribution with correction bias; all the remaining algorithms are denoted using bibliographic citations.

An advantage of the pseudo PML approach is that it one can use any algorithm to compute the part corresponding to the PML estimate as a black box. In Figure 3, we perform additional experiments for six different distributions comparing the PML estimate computed using our algorithm ("Our work") versus the algorithm in [PJW17] ("Pseudo-PJW17"), a heuristic approach to compute the approximate PML distribution.

Figure 2: Experimental results for entropy estimation.

Figure 3: Experimental results for entropy estimation.

In the remainder we provide further details on the implementation of our algorithm (Algorithm 2). In Step 1, we use CVX[GB14] with package CVXQUAD[FSP17] to solve the convex program. The accuracy of discretization determines the number of variables in the convex program and for practical purposes we perform very coarse discretization which reduces the number of variables to our convex program and helps implement Step 1 faster. The size of the discretization set we choose is slightly more than the number of distinct frequencies. Even with such coarse discretization, we still achieve results that are comparable to the other state-of-the-art entropy estimators. The intuition behind to choice of such a discretization set is because of Lemma 4.3, which guarantees the existence of a sparse solution. As the discretization set is already of small size, we do not require to perform further scarification and we avoid invoking the Sparse subroutine; therefore providing a faster practical implementation.