[Reviews · NeurIPS 2020]

Review 1

Summary and Contributions: Statistical property estimation is an important and active area at the intersection of theoretical computer science, statistics, and information theory. For example, a basic question in this realm: given n iid samples from an unknown discrete distribution p, how well can we estimate the entropy H(p), and what is an efficient algorithm for doing so? Recent efforts have shown that, for any symmetric property, the profile maximum likelihood estimator is universally minimax optimal for a wide range of parameters. While this at first seemed like a purely theoretical result, algorithmic efforts quickly caught up to show that 1) efficient approximation of the profile maximum likelihood estimator is possible and 2) approximate profile maximum likelihood estimation suffices for minimax optimality. In this context, this paper refines recent approximation algorithms from exp(-\sqrt{n} log n) to exp(-k log n) where k is the number of observed frequencies, with k = O(\sqrt{n}). This is accomplished by drawing upon graph-theoretical arguments to improve upon approximation guarantees from existing work. These bounds are complemented by numerical experiments that show that the performance of the proposed estimator is comparable to prior art.

Strengths: From a technical standpoint, this is an impressive paper that draws upon deep facts from graph theory to refine the sparsification and matrix rounding steps in the approximation algorithm. Combining these improvements with the prior work from Anari et al. 2020, the authors are able to obtain an exp(-k log n) approximation. I think this is important work towards a full picture of symmetric property estimation.

Weaknesses: Edit after author response: I am satisfied with the authors' response regarding the issues raised below and I have increased my overall score accordingly. I think the biggest gap in the paper is that it does not convincingly show that the regime k << \sqrt{n} is important with respect to numerical experiments. In particular, all of the provided plots show approximation error that is comparable to prior work, but they do not demonstrate any advantage with respect to runtime or other parameters. Therefore, it remains unclear what practical advantage is provided by this refined algorithm. Note that it very well may be the case that the algorithms from prior work have similar performance, but it is difficult to prove that this is the case. Some further discussion by the authors would be appreciated.

Correctness: To the best of my understanding, the claims of the paper and the numerical experiments are correct.

Clarity: The paper is very well-written.

Relation to Prior Work: The paper is very carefully placed within the context of prior work. It draws upon prior results where helpful, and does not exaggerate its own contributions relative to those from prior work, which is much appreciated. As a result, the contribution of this paper in terms of the approximation guarantee is very easy to understand.

Reproducibility: Yes

Additional Feedback:


Review 2

Summary and Contributions: This paper presents a new efficient algorithm for approximately computing the PML distribution. The proposed method matches the best known efficient algorithms for computing approximate PML and improves when the number of distinct observed frequencies is small. This work obtains the first computationally efficient implementation of PseudoPML. The authors also conduct experiments to evaluate their method.

Strengths: Theoretically, the authors prove that the proposed algorithm computes an exp(k log n) approximate PML distribution where k is the number of distinct observed frequencies ( k \leq sqrt{n} ), generalizing the best known result from that computed an exp( sqrt{n} log n)-approximate PML. This paper exploits interesting sparsity structure in convex relaxations of PML and provides a novel matrix rounding algorithm. The authors provide a simplified instantiation of their results that is sufficient for implementing PseudoPML, which is believed to be a key step towards practical PseudoPML.

Weaknesses: The main concern about this paper is that it seems the proofs rely heavily on the lemmas and theorems from previous papers, for example [CSS19a] and [ACSS20]. The technical contribution of this paper and challenging parts of the proofs are not very clear to the readers. Is there a main proving step to obtain the factor $k$ instead of $\sqrt{n}$? The authors may need to further clarify the technical contribution in the proofs that is different from the previous works( [CSS19a] and [ACSS20] ). ======================== Thank the authors for the response. The authors address the concerns. I raise the score from 6 to 7.

Correctness: The theoretical claims look correct. The detailed proofs are not checked carefully. The empirical methodology is correct.

Clarity: The writing of this paper is good.

Relation to Prior Work: The authors have clearly compared their results with previous works.

Reproducibility: Yes

Additional Feedback:


Review 3

Summary and Contributions: Symmetric distribution properties such as support size, support coverage, entropy, and proximity to uniformity, are of prominent interest in machine learning. Profile maximum likelihood is a sample competitive approach for the estimation of such symmetric properties, which is in particular asymptotically sample-optimal for all the above properties.This paper proposes new and substantial improvements to the algorithmic side of the PLM estimation problem. New theoretical tools are introduced and the analysis is refined and deep. Some numerical results illustrate the theoretical contributions.

Strengths: The paper's contribution is worthwhile. Rigorous proofs are provided for the claimed results. A new matrix rounding is introduced and analysed in great details. The proposed approach seems very novel and powerful.

Weaknesses: I would have appreciate a wider account of the applicability of PML for practical problems.

Correctness: The proofs I have reviewed are correct.

Clarity: The paper is well written but rapidly enters into very technical details. I would appreciate if the authors could smooth the presentation out and explain the strategy in a less dry style.

Relation to Prior Work: Relationship with prior work is clearly stated.

Reproducibility: Yes

Additional Feedback:

[Author Response · NeurIPS 2020]

We thank the reviewers for the helpful comments, valuable suggestions, and positive feedback.

**Reviewer 1:** Thank you for the feedback. We are glad you appreciate our contributions towards obtaining a more
complete picture of symmetric property estimation. First, we want to reiterate why we think the regime $k \ll \sqrt{n}$ is
interesting in theory. We think this is an important regime in large part due to recent advances in PseudoPML and profile
entropy. These results provide general approaches for symmetric property estimation with provably better statistical
guarantees under various settings. Previously no provably efficient algorithms were known to implement them; our
algorithms enable efficient implementation of these results.

As for the practical considerations of the $k \ll \sqrt{n}$ regime and the experiments, you raise a good point that there are
other practical algorithms and heuristic approximate PML algorithms that may achieve comparable estimation error
(which our experiments in some cases corroborate). Nevertheless, we believe our results are novel and of interest to the
NeurIPS community due to their provable efficiency and worst case guarantees. Our experiments were mainly focused
on demonstrating that our provable method can work as well as all previous known estimators including the entropy
specific estimator [JVHW15]. Now we do not know how well some of these other heuristic and universal methods
perform in the worst case; finding worst case distribution for these other approaches which disprove their efficiency or
proving that the other heuristic methods do well, is interesting follow up work, but outside the scope of this result.

As for experiments regarding run time, we want to point out that the algorithms we compare to fall in different categories
(heuristic [PJW17], property specific [JVHW15], universal approach for constant error regime [VV11b] and provable
PML based approaches [ACSS20, current work]). It is not surprising that the algorithms in the first three categories seem
to have better run times than provable PML based approaches as they are either fine tuned for the specific property, lack
statistical guarantees, or provably work in a limited error regime. To further improve the running time of the provable
PML based approaches involves building an efficient practical solver for a specific convex optimization problem, which
we believe is an important problem in itself and outside the scope of our work. Within the category of provable PML
based approaches (that includes [ACSS20] and our current work), we believe our algorithm is faster in practice for
problems like entropy estimation. Since in our algorithm, we implement the PseudoPML approach that requires the
computation of an approximate PML distribution (the major computational bottleneck) on a smaller input size, we have
a run time advantage relative to [ACSS20], a vanilla PML based approach (a similar phenomenon was empirically
observed earlier in [CSS19b]). We will add a more detailed discussion on the run time of different approaches in the
final version of the paper.

**Reviewer 2:** Thank you for your comments. We agree that we could do better in clarifying our technical contribution.
Regarding your concern of the relation of this paper to prior work: yes, we heavily use the previous results from
[CSS19a] and [ACSS20] but there is a clear delineation between these prior results and our own. Previous results
[CSS19a] and [ACSS20] provided a convex relaxation to the PML objective but rounding the fractional solution with
the desired guarantee $(\exp(-k \log n))$ was open – this is our contribution. The rounding algorithms provided in the
prior work [CSS19a] and [ACSS20] had worst case approximation ratios lower bounded by $\exp(-n^{2/3} \log n)$ and
$\exp(-\sqrt{n} \log n)$ respectively. There were several challenges in obtaining an improved rounding algorithm to obtain an
$\exp(-k \log n)$ approximation and thereby obtaining such an improved instance based approximation guarantee. We
addressed these challenges by understanding the sparsity structure of the convex relaxation (Lemma 4.3) and further
providing a novel matrix rounding algorithm (Theorem 4.4) that draws interesting connections to graph theory. Indeed
these are the main steps in obtaining factor $k$ instead of $\sqrt{n}$ in the approximation ratio. Further, we provide a more
practical and theoretically provable rounding algorithm for the purposes of PseudoPML and show that it performs well
in experiments. We will incorporate a more detailed version of this discussion in the paper.

**Reviewer 3:** Thank you for the feedback. We are glad you appreciate our contributions and find them novel and
powerful. To smooth out the presentation, we will add an overview of techniques section in the final version and explain
the proof strategy before getting into technical details. Regarding the practical applications of PML and universal
estimators in general, we are unaware of direct immediate practical applications. However, the line of work on universal
estimators is quite recent; we hope that in the longer term, these results could possibly yield an efficient toolkit for
effectively answering a variety of statistical questions in practice, especially for new symmetric properties lacking
custom estimators. We believe the work on PML is broadly related to questions on the phenomenon of universality and
might have interesting connections with universal sketches that further have interesting practical applications. Exploring
these connections and finding practical applications of PML are very interesting questions and directions for future
work.

[Meta-Review · NeurIPS 2020]

This paper proposes new and substantial improvements to the algorithmic side of the PLM estimation problem. New theoretical tools are introduced and the analysis is refined and deep. The authors seem to have adequately addressed all of the concerns in the rebuttal.